# Delving into Semantic Scale Imbalance

**Yanbiao Ma & Licheng Jiao & Fang Liu & Yuxin Li & Shuyuan Yang & Xu Liu**
Key Laboratory of Intelligent Perception and Image Understanding of the Ministry of Education
School of Artificial Intelligence, Xidian University
Xi'an, 710071, China
{ybmamail,yxli_12}@stu.xidian.edu.cn, lchjiao@mail.xidian.edu.cn,
f63liu@163.com, syyang@xidian.edu.cn, xuliu361@163.com

## Abstract

Model bias triggered by long-tailed data has been widely studied. However, measure based on the number of samples cannot explicate three phenomena simultaneously: **(1)** Given enough data, the classification performance gain is marginal with additional samples. **(2)** Classification performance decays precipitously as the number of training samples decreases when there is insufficient data. **(3)** Model trained on sample-balanced datasets still has different biases for different classes. In this work, we define and quantify the semantic scale of classes, which is used to measure the feature diversity of classes. It is exciting to find experimentally that there is a marginal effect of semantic scale, which perfectly describes the first two phenomena. Further, the quantitative measurement of semantic scale imbalance is proposed, which can accurately reflect model bias on multiple datasets, even on sample-balanced data, revealing a novel perspective for the study of class imbalance. Due to the prevalence of semantic scale imbalance, we propose semantic-scale-balanced learning, including a general loss improvement scheme and a dynamic re-weighting training framework that overcomes the challenge of calculating semantic scales in real-time during iterations. Comprehensive experiments show that dynamic semantic-scale-balanced learning consistently enables the model to perform superiorly on large-scale long-tailed and non-long-tailed natural and medical datasets, which is a good starting point for mitigating the prevalent but unnoticed model bias. Furthermore, we look ahead to future challenges in the field.

## 1 Introduce

In practical tasks, long-tailed class imbalance is a common problem, and the imbalance in number makes the trained model easily biased towards the dominant head classes and perform poorly on the tail classes [25; 78]. However, what is overlooked is that, in addition to long-tailed data, our study finds that the model trained on sample-balanced data still shows different biases for different classes. This model bias is not taken into account by the study for class imbalance problem, and it cannot be ameliorated by the current methods proposed for long-tailed data [5; 26; 34; 69]. For natural datasets, classes artificially divided by different semantic concepts correspond to different semantic scales, which can lead to different degrees of optimization when the deep metric is a single scale [52]. In this study, we attempt to uncover more information from the data itself and introduce and quantify the semantic scale imbalance for representing more general model bias. The semantic-scale-balanced learning is further proposed, which is used to improve loss to mitigate model bias.

The classes corresponding to different semantic concepts have different feature diversity, and we equate the diversity to the semantic scale. Usually, the finer the semantic concept of a class label, the less rich the feature diversity, the less information a model can extract, and the worse the model performs on that class [11; 52; 7; 75; 9; 72]. The manifold distribution hypothesis [57] states that a specific class of natural data is concentrated on a low-dimensional manifold. The larger the range of value variation along a specific dimension of the manifold, such as illumination and angle, the richer the feature and the larger the volume of the manifold. For example, in Figure 1, since "Swan" is a subclass of "Bird", its semantic concept is finer, so the feature diversity of "Bird" is richer than that of "Swan", and the corresponding volume of manifold is larger.

Obviously, the semantic scale can be measured by the volume of manifold. We present a reliable and numerically stable quantitative measurement of the semantic scale and further define the semantic scale imbalance. To avoid confusion, we refer to the volume of the manifold calculated on the sample space as the sample volume and the feature space as the feature volume. In addition, our innovative study for semantic scale imbalance can simultaneously and naturally explain the following two phenomena that cannot be explicated by existing studies of class imbalance:

**(1)** As the number of samples increases linearly, the model performance shows a trend of rapid improvement in the early stages and leveling off in the later stages [61].

**(2)** Even models trained on the dataset with balanced sample numbers still suffer from class bias.

The experiments demonstrate that semantic scale imbalance is more widely present in natural datasets than sample number imbalance, allowing the study scope for class imbalance to be extended to arbitrary datasets. Then, how to mitigate the adverse effects of semantic scale imbalance? We are inspired by classification methods for long-tailed data, and current solutions for class imbalance problem usually adopt re-sampling strategies [59; 80; 6] and cost-sensitive learning [24; 74; 36]. However, re-sampling may either introduce a large number of duplicate samples, making the model susceptible to overfitting when oversampling, or discard valuable samples when undersampling. Therefore by drawing on the classical re-weighting strategy [54], we propose the dynamic semantic-scale-balanced learning. Its core idea is to dynamically rather than invariably measure the degree of imbalance between semantic scales in the feature space, to achieve a dynamic evaluation of the weaker classes and assign greater weights to their corresponding losses.

Figure 1: The features from "Bird" mapped by CNNs are concentrated on a low-dimensional manifold, and the three-color point sets represent the three sub-classes of "Bird". Among them, the orange point set represents "Swan", whose feature volume is obviously smaller than that of "Bird". The classification experiments on sample-balanced datasets show that the models are biased towards the classes with larger semantic scales, such as the decision surface shown by the green line. In this case, the re-weighting strategy based on the number of samples does NOT work, while our proposed re-weighting approach based on the semantic scale biases the decision surface toward the class with a larger feature volume (red line).

In this work, our **key contributions** are summarized as:

**(1)** We propose the novel idea of leveraging the volume of manifold to measure the semantic scale (Sec 3.2). It is also innovative to find that the semantic scale has the marginal effect (Sec 3.3), and that the semantic scale of the dataset is highly consistent with model performance in terms of trends.

**(2)** We introduce and define semantic scale imbalance, aiming to measure the degree of class imbalance by semantic scale rather than sample number, which reveals a new perspective for the study of class imbalance problem. Experiments show that semantic scale imbalance is prevalent in the dataset and can more accurately reflect model bias that affects model performance (Sec 3.4).

**(3)** Semantic-scale-balanced learning is proposed to mitigate model bias, which includes a general loss improvement scheme (Sec 4.1) and a dynamic re-weighting training framework (Sec 4.2) that overcomes the challenge of calculating semantic scales in real-time during iterations. Comprehensive experiments demonstrate that semantic-scale-balanced learning is applicable to a variety of datasets and achieves significant performance gains on multiple vision tasks (Sec 5).

## 2 SLOW DRIFT PHENOMENON OF FEATURES AND MARGINAL EFFECT

Since the model parameters are changing during training, [66] studies the drifting speed of embeddings by measuring the difference in features of the same instance across training iterations. Experiments on the Stanford Online Products (SOP) dataset [41] show that the features change drastically in the early stage of training, become relatively stable after traversing the dataset twice, and drift gets

extremely slowly when the learning rate decreases. This ensures that it is reasonable to leverage historical features to calculate semantic scales.

Marginal effect [11] describes that in the early stages of model training, the network is able to learn features quickly, but since there is information overlap among samples, as the number of samples increases, the information of data will gradually saturate and the improvement in model performance from newly added samples will diminish. The effective number of samples is proposed to represent the information of data, but its limitation is that it does not work when the number of samples for each class is balanced.

Assume that the volume of each sample is unit volume 1 [11] and define the set of all samples for a class as $\Omega$ with volume $N$ and $N \geq 1$. A new sample may overlap with a previous sample such that the probability of overlap is $P$ and that of non-overlap is $1-P$. As the number of samples increases, the probability $P$ will be higher.

Define the effective number of samples as $E_n$, and $E_n = 1 + \beta \frac{1-\beta^{n-1}}{1-\beta} = \frac{1-\beta^n}{1-\beta}$ [11], where $n$ denotes the number of samples, hyperparameter $\beta = \frac{N-1}{N} \in [0,1)$ controls how fast $E_n$ grows as $n$ increases. When $N = 1$, $\beta = 0$ and $E_n = 1$, meaning that all samples can be represented by a single prototype via data augmentation. When $N \to \infty$, $\beta \to 1$, implying that there is no overlapping, then $\lim_{\beta \to 1} \frac{1-\beta^n}{1-\beta} = \lim_{\beta \to 1} \frac{(1-\beta^n)'}{(1-\beta)'} = \lim_{\beta \to 1} \frac{-n\beta^{n-1}}{-1} = n$, which indicates that the effective number of samples does not increase faster than the number of samples, but this is not consistent with our observation.

The effective number of samples $E_n$ is an exponential function of the number of samples $n$. The hyperparameters $\beta$ corresponding to classes of different grain should be different. However, the selection of $\beta$ requires more information from the data itself, but this problem is not addressed by [11], which is forced to assume that $\beta$ is the same for all classes. In this case, compared to the number of samples, $E_n$ simply uses the exponential function to obtain smoother weights. Furthermore, when the number of samples is balanced, $E_n$ is the same for each class and cannot be used to mitigate model bias, so we attempt to mine the data for information (or feature diversity) of each class to facilitate the study of imbalance problem.

## 3 SEMANTIC SCALE IMBALANCE

In this section, first, sample volume, feature volume, and semantic scale imbalance are defined. Next, we derive a quantitative measurement of feature volume to measure the semantic scale from the perspective of singular value decomposition of the data matrix and information theory. Then, the marginal effect of semantic scale is investigated. Finally, we discuss the relationship between semantic scale imbalance and model bias.

### 3.1 DEFINITIONS

Different semantic concepts correspond to different semantic scales, for example, the scale of "Bird" is larger than that of "Swan". For each class, we equate its feature diversity to its semantic scale and measure the semantic scale by the volume of subspace spanned by samples or features. Deep neural networks can be viewed as a combination of a feature mapping function $f(x, \theta)$ and a trained downstream classifier $g(z)$, i.e., $x \to z = f(x, \theta) \to y = g(z)$. Let the samples of a class be $X = [x_1, x_2, \ldots, x_m]$, and the embeddings learned by deep neural networks are represented as $Z = \{z_i | z_i = f(x_i, \theta) \in \mathbb{R}^d, i = 1, 2, \ldots, m\}$.

**Definition 3.1.** (Sample volume) The volume of the subspace spanned by sample set $X$.

**Definition 3.2.** (Feature volume) The volume of the subspace spanned by feature vectors $Z$.

**Definition 3.3.** (Semantic scale imbalance) A phenomenon of imbalance in the size of semantic scales measured by sample volume or feature volume.

### 3.2 QUANTIFICATION OF SEMANTIC SCALE

Given the data $X = [x_1, x_2, \ldots, x_m]$ and the learned embeddings $Z = [z_1, z_2, \ldots, z_m] \in \mathbb{R}^{d \times m}, z_i = f(x_i, \theta) \in \mathbb{R}^d, i = 1, 2, \ldots, m$, the volume of subspace spanned by the ran-

dom vector $z_i$ (i.e., the feature volume) is derived below, and the sample volume can be calculated in the same way (Appendix E). The covariance matrix of random vector $z_i$ is estimated as $\Sigma = E\left[\frac{1}{m}\sum_{j=1}^{m} z_j z_j^T\right] = \frac{1}{m}ZZ^T \in \mathbb{R}^{d \times d}$, $\lambda_1 \geq \lambda_2 \geq \cdots \geq \lambda_d > 0$ are the eigenvalues of real symmetric matrix. The singular value decomposition (SVD) of $Z$ yields $Z = U\Sigma V^T$ and the singular values are $\sigma_j = \sqrt{\lambda_j}, j = 1, 2, \ldots, d$. The volume of the space spanned by the vector $z_i$ is proportional to the product of all singular values of the feature matrix $Z$, i.e., $Vol(Z) \propto \prod_{j=1}^{d} \sigma_j = \sqrt{\prod_{j=1}^{d} \lambda_j}$. After the determinant expansion, the characteristic polynomial of the matrix $\Sigma$ is $\Phi(\lambda) = \det(\lambda I - \Sigma) = \lambda^d - (a_{11} + a_{22} + \cdots a_{dd})\lambda^{d-1} + \cdots + (-1)^d \det\Sigma$, and $\lambda_1 \lambda_2 \cdots \lambda_d = \det\Sigma$. Therefore, the volume of the space spanned by the vector $z_i$ is proportional to the square root of the determinant of the covariance matrix of $Z$:

$$Vol(Z) \propto \sqrt{\det(\frac{1}{m}ZZ^T)}. \tag{1}$$

The same result can be derived from the volume of a parallel hexahedron defined by vectors (Appendix G). Considering that real-world metric tools typically have a dynamic range, for example, a ruler always has multiple scales (1mm, 1cm, or even 10cm) to measure objects of different scales, we expect the quantitative measurement of feature volume to have a multi-scale metric and therefore use the sphere packing method [8; 37], which is normally adopted in information theory, to implement it.

There is an error at the boundary when filling with hyperspheres because all spheres cannot be exactly tangent to the edges of the manifold. The error of the finite feature vectors is assumed to be independent additive Gaussian noise [8] : $z_i' = z_i + w_i$, where $w_i \sim N(0, \frac{\varepsilon^2}{n}I)$ (n is the space dimension). The estimate of the number of spheres of radius $\varepsilon$ needed to pack the space spanned by all vectors is $N_\varepsilon = Vol(Z')/Vol(Ball)$. The adjustment of the metric scale can be achieved by tuning the radius $\varepsilon$ of the spheres, thus controlling the measurement result of feature volume. Then the covariance matrix of the vector $z_i$ is $\Sigma' = \frac{\varepsilon^2}{n}I + \frac{1}{m}ZZ^T \in \mathbb{R}^{d \times d}$, such that $Vol(Z') \propto \sqrt{\det(\frac{\varepsilon^2}{n}I + \frac{1}{m}ZZ^T)}$ and $Vol(Ball) \propto \sqrt{\det(\frac{\varepsilon^2}{n}I)}$. The feature volume is proportional to $N_\varepsilon$:

$$Vol(Z) \propto N_\varepsilon = \frac{Vol(Z')}{Vol(Ball)} = \sqrt{\frac{\det\left(\frac{\varepsilon^2}{n}I + \frac{1}{m}ZZ^T\right)}{\det\left(\frac{\varepsilon^2}{n}I\right)}} \tag{2}$$
$$= \sqrt{\det\left(I + \frac{n}{m\varepsilon^2}ZZ^T\right)}.$$

The dimension of feature $z_i$ is $d$, so let $n = d$. In order to increase the numerical stability, we perform a logarithmic transformation of the above equation, which does not affect the monotonicity of the function, and we can obtain $Vol(Z) \propto \log_2\sqrt{\det\left(I + \frac{n}{m\varepsilon^2}ZZ^T\right)} = \frac{1}{2}\log\det\left(I + \frac{d}{m\varepsilon^2}ZZ^T\right)$. In practical training, it is essential to normalize the feature vectors so that their mean value is 0. In this work, we set $\varepsilon = 1000$, and the value of $\varepsilon$ does not affect the relative size of the space spanned by feature vectors of each class. The volume of the space spanned by $Z$ can be can be written as:

$$Vol(Z) = \frac{1}{2}\log_2\det(I + \frac{d}{m}(Z - Z_{mean})(Z - Z_{mean})^T), \tag{3}$$

where $Z_{mean}$ is the mean value of $Z$, $Vol(Z) > 0$ when the number of samples $m > 1$. We measure the semantic scale $S'$ by the feature volume, i.e., $S' = Vol(Z)$, and the larger $S'$, the richer the feature diversity, which we verify in Appendix C using multiple Stanford point cloud manifolds.

### 3.3 Marginal Effect of Semantic Scale

The marginal effect describes that the feature richness will gradually saturate as the number of samples increases, so the change of semantic scale should also conform to the marginal effect. Figure 2 illustrates that as the number of samples increases, the semantic scale $S'$ measured by sample volume gradually saturates, which indicates that the quantitative measurement of semantic scale is as

expected. In addition, the growth rate of the semantic scale varies across classes, which is determined by the grain size of the class itself. It leads to different semantic scales even if all classes have the same number of samples.

To investigate why adding samples has a marginal improvement in model performance when the training samples are sufficient, we use the following method to generate new training datasets for classification experiments: assume that the total number of classes in the original dataset is $C$, and $m$ samples are randomly selected from each class to form a sub-dataset with a total number of samples of $C \times m$. The sub-datasets generated based on CIFAR-10, CIFAR-100 [30] and Mini-ImageNet [62] are shown in Appendix D.1.

We train ResNet-18 and ResNet-34 [17] on each of the **31** training sets in Table 6, and the sum of the semantic scales for all classes and the corresponding top-1 accuracy are shown in Figure 2. We are pleasantly surprised to find that when the semantic scale increases rapidly, the model performance improves swiftly with it, and when the semantic scale becomes saturated, the improvement is small.

## 3.4 SEMANTIC SCALE IMBALANCE AND MODEL BIAS

### 3.4.1 QUANTIFICATION OF SEMANTIC SCALE IMBALANCE

Table 1: Pearson correlation coefficients between the accuracy of classes and the semantic scales $S$ with different $\alpha$. $N$ denotes the number of samples, and $S'$ represents the semantic scale without considering inter-class interference. $E_n$ denotes the number of effective samples.

| Dataset | Model | $N$ | $E_n$ | $W$ | $S'$ | $S$ | | | | |
|---|---|---|---|---|---|---|---|---|---|---|
| | | | | | | $\alpha=1$ | $\alpha=1.5$ | $\alpha=2$ | $\alpha=2.5$ | $\alpha=3$ |
| CIFAR-10-LT | ResNet-18 | 0.8346 | 0.8664 | 0.2957 | 0.8688 | 0.8456 | **0.9603** | 0.9553 | 0.9398 | 0.9269 |
| | ResNet-34 | 0.7938 | 0.8476 | 0.3186 | 0.9426 | 0.7950 | 0.9678 | **0.9884** | 0.9854 | 0.9796 |
| CIFAR-10 | ResNet-18 | 0.0950 | 0.0950 | 0.1743 | 0.5433 | **0.7850** | 0.7250 | 0.6644 | 0.6060 | 0.5607 |
| | ResNet-34 | 0.1502 | 0.1502 | 0.2075 | 0.5750 | **0.8056** | 0.7442 | 0.6870 | 0.6465 | 0.5906 |

The previous subsection shows that the sum of semantic scales for all classes in the dataset is highly correlated with the model performance, and we further investigate the relationship between semantic scale and model bias for different classes. When a class is closer to other classes, the model performs worse on that class [40; 1]. Therefore, inter-class interference is additionally considered when quantifying the degree of imbalance between semantic scales of classes. When class $i$ is closer to other classes, a smaller weight $w_i$ is applied to the semantic scale of class $i$.

Specifically, the semantic scale of $m$ classes after maximum normalization is assumed to be $S' = [S'_1, S'_2, \ldots, S'_m]^T$, and the centers of all classes are $O = [o_1, o_2, \ldots, o_m]^T$. Define the distance between the centers of class $i$ and class $j$ as $d_{i,j} = \|o_i - o_j\|_2$, the weight $w_i = \frac{1}{m-1} \sum_{j=1}^{m} \|o_i - o_j\|_2$. The weights of $m$ classes are written as $W' = [w_1, w_2, \ldots, w_m]^T$. After the maximum normalization and logarithmic transformation of $W'$, we can obtain $W = \log_2(\alpha + W'), \alpha \geq 1$, where $\alpha$ is used to control the smoothing degree of $W$. After considering the inter-class distance, the semantic scale $S = S' \odot W$, and the role of $S'$ in dominating the degree of imbalance is greater when $\alpha$ is larger.

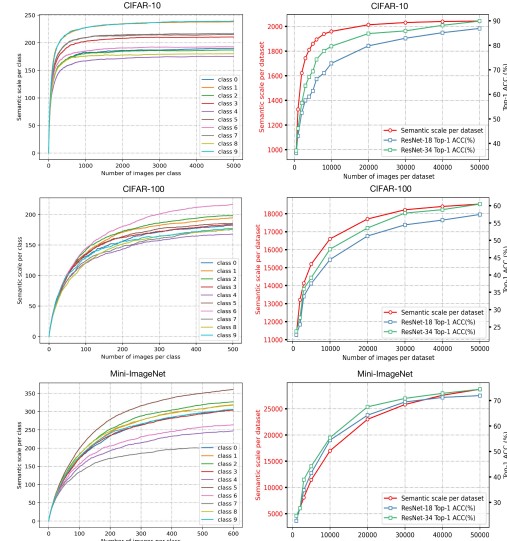

Figure 2: **Left column**: curves of semantic scales with increasing number of samples for the first ten classes from different datasets. **Right column**: for different sub-datasets, curves of the sum of semantic scales for all classes and top-1 accuracy curves of trained ResNet-18 and ResNet-34. All models are trained using the Adam optimizer [28] with an initial learning rate of 0.01 and then decayed by 0.98 at each epoch.

To obtain the most appropriate $\alpha$, we calculate the Pearson correlation coefficients between the semantic scale and the accuracy of ResNet-18 and ResNet-34 trained on CIFAR-10-LT and CIFAR-

10, as shown in Table 1. The experimental settings are in Appendix D.2. It can be found that $S'$ is more dominant on long-tailed data than on non-long-tailed data, and the improved $S$ is better than $S'$ and far better than the number of samples in reflecting model bias. In the following experiments, we let $\alpha$ be 2 on the long-tailed data and 1 on the non-long-tailed data.

### 3.4.2 SEMANTIC SCALE IMBALANCE ON LONG-TAILED DATA

Previous studies have roughly attributed model bias to the imbalance in the number of samples. The experimental results in the first row of Figure 3 show that even though the number of MNIST-LT-1 is similar to that of MNIST-LT-2, the class-wise accuracy on MNIST-LT-1 is closer to that on MNIST, just as their semantic scales are also more similar.

In addition, Figure 3 indicate that models on certain classes with fewer samples outperform those on classes with more samples, and that the semantic scale $S$ reflects model bias more accurately. Further, we also observe that the accuracy of the CIFAR-100-LT does not show a significant decreasing trend, which can be explained by the marginal effect of the semantic scale (Sec 3.3).

### 3.4.3 SEMANTIC SCALE IMBALANCE ON NON-LONG-TAILED DATA

Figure 4 demonstrates the model bias not only on long-tailed data but also on sample-balanced data. Usually, the classes with smaller semantic scales have lower accuracies. Depending on the size of the semantic scale, it can make it possible for the weaker and dominant classes to be well differentiated. It should be noted that the weaker classes are not random, and experiments in Figure 4 show that the models always perform worse on the same classes. More semantic scale imbalance of the datasets is shown in Appendix D.4. In summary, semantic scale imbalance can represent model bias more generally and appropriately, and further, we expect to improve the overall performance of the model when facing the semantic scale imbalance problem. Therefore, we propose the dynamic semantic-scale-balanced learning by drawing on the re-weighting strategy.

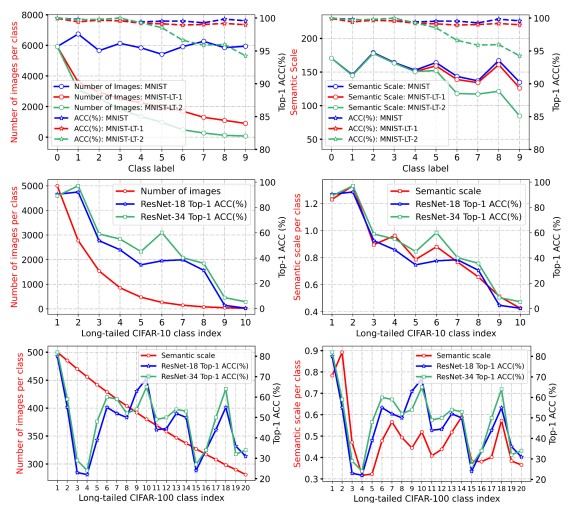

Figure 3: Correlation study of accuracy with the number of samples and semantic scale on MNIST, MNIST-LT (Appendix D.3), CIFAR-10-LT and CIFAR-100-LT datasets.

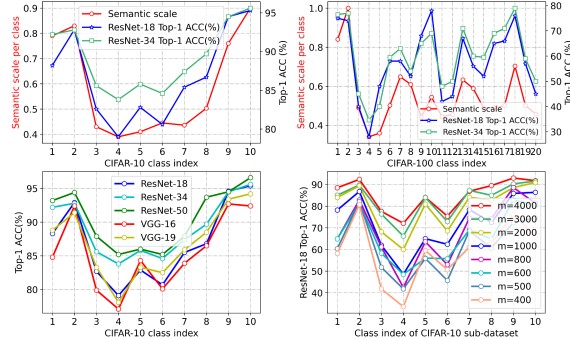

Figure 4: **Top row**: correlation study between accuracy and semantic scale on the sample-balanced dataset. **Bottom row**: performance of different models [48] trained on the CIFAR-10 dataset and performance of ResNet-18 trained on different sub-datasets of CIFAR-10 from Table 6 of Appendix D.1.

## 4 DYNAMIC SEMANTIC-SCALE-BALANCED LEARNING

Deep neural networks can be viewed as a combination of the feature mapping function and the classifier, and several studies have shown that model bias is mainly caused by classifier bias [80; 67; 3], so we are more concerned with semantic scale imbalance in the feature space, i.e., semantic scale measured by the **feature volume**. In this section, we propose a general semantic-scale-based loss improvement scheme and design a training framework for the successful application of the scheme.

### 4.1 DYNAMIC SEMANTIC-SCALE-BALANCED LOSS

During training, the feature vectors corresponding to the samples vary with the model parameters, and thus the semantic scale per class is constantly changing. Compared with the traditional re-weighting

strategy, we propose to calculate the degree of imbalance between semantic scales in real-time at each iteration in order to dynamically evaluate the weaker classes and assign greater weights to their corresponding losses. Specifically, for class $i$ at each iteration, normalized re-weighting terms $\alpha_i \propto \frac{1}{S_i}, \sum_{i=1}^{C} \alpha_i = 1$, inversely proportional to the semantic scales that take into account inter-class interference, are introduced, and $C$ is the total number of classes. Given the embedding $z$ of a sample and label $y_i$, the dynamic semantic-scale-balanced (**DSB**) loss can be expressed as $DSB(z, y_i) = \frac{1}{S_i} L(z, y_i), i = 1, 2, \ldots, C$, where $y_i$ is the label of the sample from class $i$. How to combine general loss to generate DSB loss is described in Appendix F.1. Our approach has great potential to improve the methods of re-balancing loss and adjusting sampling rate based on the number of samples, because both the semantic scale and the number of samples are natural measures and they are not model-dependent.

However, the number of samples used at each iteration is limited, and it is not possible to obtain the features of all samples for calculating the semantic scales. Therefore, we propose a dynamic re-weighting training framework that enables DSB loss to be successfully applied.

## 4.2 Dynamic Re-Weighting Training Framework

Inspired by the slow drift phenomenon of features [66; 35], we design a storage pool $Q$ to store and update historical features and propose a three-stage training framework. A mini-batch of features can be dynamically updated at each iteration, and the semantic scale of each class is calculated using all the features in the storage pool. The three-stage training framework is shown in Figure 16 and Algorithm 2 (More details are in Appendix F.2), with the following textual description.

**(1)** In the **first stage**, all the features and labels generated by the 1st epoch are stored in $Q$, but they cannot be used directly to calculate semantic scales due to the large drift of historical features from current features in the early stage.

**(2)** The **second stage** corresponds to epoch 2 to epoch $n$. At each iteration, the oldest mini-batch features and labels in $Q$ are removed and those generated by the current iteration are stored. The goal is to continuously update the features in $Q$ until the feature drift is small enough. We set $n$ to 5 in our experiments, and the original loss function is used in the first two stages. Figure 5 shows the effect of $n$ on the model performance. A larger $n$ does not hurt the model performance, but only takes a little more time. Experience suggests that setting $n$ to 5 is sufficient.

**(3)** The **third stage** corresponds to epoch $> n$. At each iteration, the semantic scales are calculated using the features in $Q$ after updating $Q$, and the original loss is re-weighted.

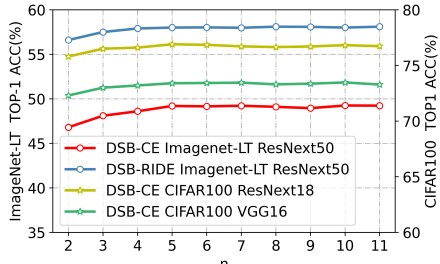

Figure 5: The performance of different models with different losses on different datasets for different values of $n$.

The comparison and analysis of the **video memory** and **training speed** are in Appendix F.2. We answer possible questions about the methods section in detail in Appendix B.

## 5 Experiments

To validate the superiority and generality of the proposed dynamic semantic-scale-balanced learning, we design four experiments. The **first experiment** is conducted on large-scale long-tailed datasets, ImageNet-LT and iNaturalist2018 [60], to confirm the superior performance of our approach on long-tailed data. The **second experiment** uses large-scale ImageNet [45] and sample-balanced CIFAR-100, and the **third experiment** selects CIFAR-100-LT and benchmark datasets commonly used in deep metric learning (CUB-2011 [63] and Cars196 [29]). The **fourth experiment** is performed on MSCOCO-GLT [56] to demonstrate the effectiveness of our approach in generalized long-tailed classification. The comprehensive experiments demonstrate the generality and superiority of our proposed method. More experimental results are provided in Appendix D.5, Appendix D.6 (**Results on the fundus dataset OIA-ODIR** [31]) and Appendix D.7 (**Remote sensing image scene classification**). The ablation experiments and additional analyses are in Appendix H.

Table 2: Top-1 Acc(%) on ImageNet-LT and iNaturalist2018. We use ResNext-50 [70] on ImageNet-LT and ResNet-50 [17] on iNaturalist2018 as the network backbone for all methods. And we conduct model training with the SGD optimizer based on batch size 256 (for ImageNet-LT) / 512 (for iNaturalist), momentum 0.9, weight decay factor 0.0005, and learning rate 0.1 (linear LR decay).

| Methods | ImageNet-LT(ResNeXt50) | | | | iNaturalist 2018(ResNet50) | | | |
|---|---|---|---|---|---|---|---|---|
| | Head | Middle | Tail | Overall | Head | Middle | Tail | Overall |
| BBN [80] | 43.3 | 45.9 | 43.7 | 44.7 | 49.4 | 70.8 | 65.3 | 66.3 |
| DIVE [18] | 64.1 | 50.4 | 31.5 | 53.1 | 70.6 | 70 | 67.6 | 69.1 |
| CE | 65.9 | 37.5 | 7.70 | 44.4 | 67.2 | 63.0 | 56.2 | 61.7 |
| CB-CE [11] | 39.6 | 32.7 | 16.8 | 33.2 | 53.4 | 54.8 | 53.2 | 54.0 |
| **DSB-CE** | **67.3** | **42.5** | **21.4(+13.7)** | **49.2(+4.8)** | **68.5** | **63.4** | **62.7(+6.5)** | **64.3(+2.6)** |
| **DSB-CE+IFL** [56] | **68.1** | **43.4** | **22.5(+14.8)** | **50.1(+5.7)** | **69.1** | **64.3** | **63.4(+7.2)** | **65.0(+3.3)** |
| Focal [11] | 67.0 | 41.0 | 13.1 | 47.2 | - | - | - | 61.1 |
| CB-Focal [11] | - | - | - | - | - | - | - | 61.2 |
| **DSB-Focal** | **68.1** | **44.2** | **23.7(+10.6)** | **50.6(+3.4)** | **70.6** | **62.8** | **58.4** | **63.5(+2.4)** |
| LDAM [4] | 60.0 | 49.2 | 31.9 | 51.1 | - | - | - | 64.6 |
| **DSB-LDAM** | **60.7** | **50.5** | **33.4(+1.5)** | **52.3(+1.2)** | **69.4** | **66.5** | **61.9** | **65.7(+1.1)** |
| BS [43] | 62.4 | 47.7 | 32.1 | 51.2 | 60.1 | 51.4 | 46.7 | 53.2 |
| **DSB-BS** | **63.2** | **48.9** | **35.4(+3.3)** | **52.8(+1.6)** | **61.4** | **52.8** | **49.4(+2.7)** | **55.1(+1.9)** |
| LADE [19] | 62.3 | 49.3 | 31.2 | 51.9 | - | - | - | 69.7 |
| **DSB-LADE** | **62.6** | **50.4** | **33.6(+2.4)** | **53.2(+1.3)** | **72.3** | **70.7** | **65.8** | **70.5(+0.8)** |
| PaCo [10] | 63.2 | 51.6 | 39.2 | 54.4 | 69.5 | 72.3 | 73.1 | 72.3 |
| DSB-PaCo | 64.1 | 52.9 | 41.5(+2.3) | 55.9(+1.5) | 70.2 | 73.4 | 74.6 | 73.4(+1.1) |
| MBJ [35] | 61.6 | 48.4 | 39.0 | 52.1 | - | - | - | 70.0 |
| **DSB+MBJ** | **63.2** | **49.6** | **40.7(+1.7)** | **53.3(+1.2)** | **73.6** | **70.2** | **66.2** | **70.9(+0.9)** |
| RIDE [65] | 67.9 | 52.3 | 36.0 | 56.1 | 70.9 | 72.4 | 73.1 | 72.6 |
| MBJ+RIDE [35] | 68.4 | 54.1 | 37.7 | 57.7 | - | - | - | 73.0 |
| **DSB+RIDE** | **68.6** | **54.5** | **38.5(+2.5)** | **58.2(+2.1)** | **70.7** | **74.0** | **74.2(+1.1)** | **73.4(+0.8)** |

## 5.1 RESULTS ON IMAGENET-LT AND INATURALIST2018

ImageNet-LT is a long-tailed version of ImageNet containing 1,000 classes with between 1,280 and 5 samples per class. iNaturalist2018 is a real-world, extremely unbalanced dataset containing 437,513 images from 8,142 classes. We adopt the official training and validation splits [11].

Table 2 shows that when CE, Focal [33] and RIDE are combined with our approach (Appendix D.1), the model overall performance is significantly improved. For example, the overall accuracy of DSB-CE is **4.8%** and **2.6%** higher than CE on ImageNet-LT and iNaturalist2018, respectively. We also report the performance on three subsets (Head: more than 100 images, Middle: 20-100 images, Tail: less than 20 images) of these two datasets. It can be observed that our proposed method has the largest improvement for the tail subset without compromising the performance of the head subset, where DSB-CE and DSB-Focal improve **13.7%** and **10.6%**, respectively, over the original method in the tail subset of ImageNet-LT, effectively alleviating the model bias. In addition, IFL [56] considers the intra-class long-tailed problem, and when we combine DSB-CE with it (i.e., DSB-CE-IFL), the performance of the model is further enhanced. Therefore, we encourage researchers to focus on the intra-class long-tailed problem.

## 5.2 RESULTS ON IMAGENET AND CIFAR-100

Table 3: Comparison on ImageNet and CIFAR-100. On ImageNet, we use random clipping, mixup [77], and cutmix [76] to augment the training data, and all models are optimized by Adam with batch size of 512, learning rate of 0.05, momentum of 0.9, and weight decay factor of 0.0005. On CIFAR-100, we set the batch size to 64 and augment the training data using random clipping, mixup, and cutmix. An Adam optimizer with learning rate of 0.1 (linear decay), momentum of 0.9, and weight decay factor of 0.005 is used to train all networks.

| Methods | ImageNet Top-1 Acc(%) | | | CIFAR-100 Top-1 Acc(%) | | |
|---|---|---|---|---|---|---|
| | CE | DSB-CE | Δ | CE | DSB-CE | Δ |
| VGG16 [48] | 71.6 | 72.9 | +1.3 | 71.9 | 73.4 | +1.5 |
| BN-Inception [53] | 73.5 | 74.4 | +0.9 | 74.1 | 75.2 | +1.1 |
| ResNet-18 | 70.1 | 71.2 | +1.1 | 75.6 | 76.9 | +1.3 |
| ResNet-34 | 73.5 | 74.3 | +0.8 | 76.8 | 77.9 | +1.1 |
| ResNet-50 | 76.0 | 76.8 | +0.8 | 77.4 | 78.3 | +0.9 |
| DenseNet-201 [22] | 77.2 | 78.1 | +0.9 | 78.5 | 79.7 | +1.2 |
| SE-ResNet-50 [21] | 77.6 | 78.4 | +0.8 | 78.6 | 79.3 | +0.7 |
| ResNeXt-101 [70] | 78.8 | 79.7 | +0.9 | 77.8 | 78.8 | +1.0 |

Table 4: Results on CUB-2011 and Cars196. We evaluate the model performance with Recall@K [41] and Normalized Mutual Information (NMI) [46].

| | Dataset | CUB-2011 | | | Cars196 | | |
|---|---|---|---|---|---|---|---|
| | Metric | R@1 | R@2 | NMI | R@1 | R@2 | NMI |
| dim 64 | NormSoftmax | 57.8 | 70.0 | 65.3 | 76.8 | 85.6 | 66.7 |
| | **DSB-NSM** | **59.2(+1.4)** | **70.7(+0.7)** | **66.5 (+1.2)** | **77.9(+1.1)** | **86.4(+0.8)** | **67.8(+1.1)** |
| | SoftTriple | 60.1 | 71.9 | 66.2 | 78.6 | 86.6 | 67.0 |
| | **DSB-ST** | **61.3 (+1.2)** | **72.7(+0.8)** | **67.3(+1.1)** | **79.8(+1.2)** | **87.5(+0.9)** | **68.3 (+1.3)** |
| dim 512 | Circle | 66.7 | 77.4 | - | 83.4 | 89.8 | - |
| | NormSoftmax | 63.9 | 75.5 | 68.3 | 83.2 | 89.5 | 69.7 |
| | **DSB-NSM** | **65.1(+1.2)** | **76.3(+0.8)** | **69.2(+0.9)** | **84.0(+0.8)** | **90.2(+0.7)** | **70.9(+1.2)** |
| | SoftTriple | 65.4 | 76.4 | 69.3 | 84.5 | 90.7 | 70.1 |
| | **DSB-ST** | **66.4(+1.0)** | **77.0(+0.6)** | **70.6(+1.3)** | **85.6(+1.1)** | **91.3(+0.6)** | **71.1(+1.0)** |

We use the ILSVRC2012 split contains 1,281,167 training and 50,000 validation images. Each class of CIFAR-100 contains 500 images for training and 100 images for testing. The results in Table 3 indicate that our approach is able to achieve performance gains greater than **1%** for a variety of networks on both datasets. In particular, it enables VGG16 to improve **1.3%** and **1.5%** on ImageNet and CIFAR-100, respectively, compared to the original method. This implies that there is a semantic scale imbalance in non-long-tailed datasets and it affects the model performance.

## 5.3 RESULTS ON CUB-2011, CARS196 AND CIFAR-100-LT

Since we also improve on the classical losses (NormSoftmax and SoftTriple [42]) in the field of deep metric learning, we abide by the widely adopted backbone network, experimental parameters, and the division of datasets in this field (Appendix D.5). The two improved loss functions are denoted as DSB-NSM and DSB-ST, respectively, and their formulas are given in Appendix F.1.

**Results on CUB-2011 and Cars196**. Table 4 summarizes the performance of our method with 64 and 512 embeddings, respectively. The experiments show that DSB loss is able to consistently improve by more than **1%** on R1 and NMI. DSB-ST with 512 embeddings performs superiorly on Cars196, where R@1 and R@2 exceed the Circle loss [51] by **2.2%** and **1.5%**, respectively.

Table 5: Results on CIFAR-100-LT. The imbalance factor of a dataset is defined as the value of the number of training samples in the largest class divided by that in the smallest class.

| | Dataset | CIFAR-100-LT | | | | | | | | |
|---|---|---|---|---|---|---|---|---|---|---|
| | Imbalance factor | 10 | | | 50 | | | 200 | | |
| | Metric | R@1 | R@2 | NMI | R@1 | R@2 | NMI | R@1 | R@2 | NMI |
| dim 64 | NormSoftmax | 54.6 | 65.2 | 62.4 | 49.6 | 60.5 | 58.0 | 43.4 | 54.5 | 52.9 |
| | CB-NSM | 55.7 | 66.1 | 63.3 | 50.5 | 61.1 | 58.7 | 45.5 | 55.3 | 53.8 |
| | **DSB-NSM** | **56.3** | **66.7** | **63.5** | **51.3** | **61.4** | **59.1** | **46.0** | **56.1** | **54.4** |
| | SoftTriple | 56.6 | 67.6 | 63.9 | 49.5 | 61.0 | 58.3 | 46.6 | 57.8 | 55.4 |
| | CB-ST | 58.1 | 68.4 | 65.1 | 51.1 | 62.8 | 59.4 | 48.2 | 59.5 | 56.6 |
| | **DSB-ST** | **58.8** | **69.0** | **65.8** | **51.5** | **62.5** | **59.7** | **49.3** | **60.6** | **57.3** |

**Results on CIFAR-100-LT**. Class-balanced loss (CB loss) that performs well on long-tailed data and is also based on the re-weighting strategy is selected for comparison with DSB loss. Analyzing the results in Table 5, the DSB loss outperforms the CB loss overall. Among them, when the imbalance factor of long-tailed CIFAR-100 is 200, DSB-ST performs significantly better than CB-ST, with higher performance than SoftTriple on R@1, R@2 and NMI by **2.7%**,**2.8%** and **1.9%**.

Extensive experiments show that the semantic scale imbalance needs to be paid extensive attention.

## 6 DISCUSSION

In this work, we pioneer the concept and quantitative measurement of semantic scale imbalance, and make two important discoveries: **(1)** semantic scale has marginal effects, and **(2)** semantic scale imbalance can accurately describe model bias. It is important to note that **our proposed semantic scale, like the number of samples, is a natural measure of class imbalance and does not depend on the model's predictions** (See Related Work in Appendix A). Semantic scale can guide data augmentation, e.g., semantic scale imbalance can evaluate which classes are the weaker classes that need to be augmented, and marginal effects can assist us to select a more appropriate number of samples. We expect that our work will bring more attention to the more prevalent model bias, improve the robustness of models and promote the development of fairer AI.

## 7 ACKNOWLEDGEMENTS

This work was supported in part by the Key Scientific Technological Innovation Research Project by Ministry of Education, the State Key Program and the Foundation for Innovative Research Groups of the National Natural Science Foundation of China (61836009), the Major Research Plan of the National Natural Science Foundation of China (91438201, 91438103, and 91838303), the National Natural Science Foundation of China (U22B2054, U1701267, 62076192, 62006177, 61902298, 61573267, 61906150, and 62276199), the 111 Project, the Program for Cheung Kong Scholars and Innovative Research Team in University (IRT 15R53), the ST Innovation Project from the Chinese Ministry of Education, the Key Research and Development Program in Shaanxi Province of China(2019ZDLGY03-06), the National Science Basic Research Plan in Shaanxi Province of China(2022JQ-607), the China Postdoctoral fund(2022T150506), the Scientific Research Project of Education Department In Shaanxi Province of China (No.20JY023), the National Natural Science Foundation of China (No. 61977052).

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

## A    RELATED WORK

Real-world datasets tend to long-tailed. The extreme imbalance in the number of samples for long-tailed data prevents the classification model from learning the distribution of the tail classes adequately, which leads to poor performance on the tail classes. Therefore, the methods of re-balancing the number of samples [26; 16; 13; 79] and balancing the loss incurred per class [12; 81; 50], i.e., re-sampling and cost-sensitive learning, are proposed. Among them cost-sensitive learning is most relevant to our work.

[44] proposes to use the frequency of labels to adjust the loss during training to mitigate class bias. [33] assigns weights to the loss for each class, and the hard samples are given higher weights. Recent studies have shown that re-weighting the loss strictly by the inverse of the number of samples is moderate [38; 39]. Some methods that generate weights for re-weighting in a more "smooth" manner perform better, such as taking the square root of the number of samples [38] as weights. CB loss [11] attributes the better performance of this more "smooth" approach to the presence of marginal effects, while other studies [54; 20; 64] attribute it to the negative gradient over-suppression. Distribution-balanced loss [68] proposes negative-tolerant regularization to mitigate gradient differences, and recalculates the loss by calculating the ratio of the expected to the actual sampling frequency for each class. Seesaw Loss [64] leverages mitigation and compensation factors to dynamically suppress excessive negative sample gradients on the tail classes while complementing the penalty for misclassified samples. Furthermore, with the proposal of decoupled training [80], [26] adopts decoupled training, using cross-entropy loss to learn features in the first stage and re-balancing loss to learn the classifier in the second stage.

Unlike the above studies of re-balancing loss, which all re-balance loss based on the number of samples or the ratio of positive and negative gradients, our work proposes a novel measure, called semantic scale. Compared to the number of samples, the semantic scale also considers the sample distribution scope. In contrast to gradient-based measures, the semantic scale does not depend on the model output and gradient back-propagation, and is a natural measure similar to the number of samples. Work on model robustness has focused on the out-of-domain generalization performance of models, an area known as "out-of-distribution generalization of models". For example, [49] aims to maintain the good performance of the model when the test distribution deviates from the training distribution. Similarly, [32] aims to allow the model to learn more information outside the domain. Unlike them, we are concerned with the problem that model bias introduced by unbalanced data makes the model perform poorly in certain classes.

## B    EXPLANATION OF A FEW KEY POINTS

### B.1    HOW DOES THE SECTION "SLOW DRIFT PHENOMENON AND MARGINAL EFFECTS OF CHARACTERISTICS" RELATE TO THE REST OF THE PAPER?

(1) Why do we have to introduce marginal effects?

The effective number of samples discusses the relationship between the sample number and feature diversity in a class, and it argues that feature diversity has marginal effects. However, the effective number of samples has many major drawbacks (introduced in Section 2), such as it does not work on sample-balanced datasets and does not give a quantitative measure of feature diversity. Therefore, we extend the mechanism of the efficient number of samples and propose the "semantic scale" that can effectively measure feature diversity in a sample-balanced dataset. On the one hand, our approach is more general compared to the effective number of samples. On the other hand, the marginal effect proves that our extension is reasonable and appropriate. In addition, our approach simultaneously explains three phenomena that cannot be explicated by other methods, which indicates the reliability of our proposed method. In brief, the logic of our paper is as follows.

- CB loss introduced the concept of the effective number of samples based on marginal effects.

- We extend the effective number of samples and propose the concept of semantic scale.

- Experiments show that the semantic scale still has marginal effects (Section 3.3).

- According to the properties mentioned in step 3, we can explore many applications based on semantic scale that require marginal effects as theoretical support (e.g., the selection method of representative samples supplemented in Appendix I).

In summary, we would like to explain that the birth of semantic scale measurement (or semantic scale imbalance based on semantic scale) was inspired by the effective number of samples with marginal effects. If the marginal effect is discarded, then our early motivation and the later practical applications are theoretically weak and unconvincing.

(2) Association with feature slow drift

Since experiments show a very high correlation between semantic scale and model bias, we propose to re-weight the loss function with the inverse of the semantic scale. The features are dynamically changing during training, and all the feature vectors are needed to calculate the semantic scale of each class. Obviously, the data in one batch is not enough, so we propose to dynamically update and store the historical features to calculate the semantic scale, and the feature slow drift phenomenon ensures the feasibility of this operation.

Section 2 is indispensable for the whole paper, it ensures the coherence of the paper.

### B.2    WHY FOCUS ON THE RELATIONSHIP BETWEEN SEMANTIC SCALE AND ACCURACY?

It is important to note that we are concerned with the model bias introduced by unbalanced data, which causes models to perform poorly on some classes. In the past, researchers believed that models performed poorly on classes with fewer samples and therefore defined classes with fewer samples as tail classes and classes with more samples as head classes, proposing a long-tailed identification task. However, we observe that the model does not necessarily perform poorly on classes with fewer samples, which explains why some of the tail classes are "overbalanced" in many long-tailed identification methods. The higher similarity between our proposed semantic scale and model performance allows us to redefine the imbalance problem by replacing the number of samples with semantic scale. The superior performance achieved on the sample-balanced datasets shows that our proposed semantic scale imbalance is reliable. The semantic scale measure does not depend on the model and is calculated directly from the data. Even more surprising is that the semantic scale of the class can predict the performance of the class, which can lead to further understanding of what the model learns from the data. It can facilitate the development of data-driven artificial intelligence.

### B.3    WHY SHOULD THE LOSS FUNCTION BE DYNAMICALLY WEIGHTED?

The working process of DSB loss: in each iteration, the semantic scale of each class is calculated in the feature space, and the loss function is re-weighted by the inverse of the semantic scale.

Why is the loss "dynamically" weighted? Because the semantic scales change continuously as the features change during training, and we need to update the features in each iteration and re-calculate the semantic scales to re-weight the loss function. The term "dynamic" refers to the dynamic update of the semantic scale in each iteration.

However, there is difficulty in implementing dynamic weighting, i.e., all features are needed to calculate the semantic scale, and we cannot extract the features of all samples in each iteration, which would be time consuming. Therefore, we analyze the "feature slow drift" phenomenon in Section 2 and propose to calculate the semantic scale by dynamically storing and updating the historical features (i.e., dynamic re-weighting training framework). Comparative experiments on the training speed and memory consumption of the above training framework are presented in Appendix F.2, and the results show that our approach is efficient.

### B.4    WHAT IS THE POINT OF PROPOSING A VARIETY OF COST-SENSITIVE LEARNING METHODS? WHY NOT DIRECTLY USE THE ACCURACY OF EACH CLASS TO WEIGHT THE LOSS?

What is the point of proposing a variety of cost-sensitive learning methods? For example, using the inverse of the number of samples  the effective number of samples to reweight the loss, rather than directly using the accuracy of each class to weight. After careful consideration, we believe there are several reasons.

(1) Reweighting loss with class accuracy may cause the model to over-focus on weak classes, so that other classes are ignored. Recent studies have shown that reweighting the loss strictly by the inverse of the number of samples has a modest effect [38; 39]. Some "smoother" methods perform better, such as taking the square root of the number of samples [38] as the weight. [11] argues that the reason why the "smoother" method performs better is due to the existence of marginal effects. Our approach can be understood as a smoothed version of class accuracy because our proposed semantic scale has marginal effects and a high correlation with class accuracy. We note a recent work (CDB loss) published in IJCV that measures class difficulty. In addition, domain balancing also measures class-level difficulty, so we compare semantic-scale-balanced learning with them. The introduction and comparison experiments of the above two methods are shown in Appendix H.2.

(2) The method of weighting with model performance cannot bring us new cognition. Why do models perform poorly on some data and well on others? For example, face recognition models usually do not perform well in dark environments. When we encounter such a problem, the first thing to think about is whether the lack of data in the dark environment causes the model to not be fully learned. Since there is a lot of data available, this problem is not caused by the few samples, so is there any other explanation? We argue that the pattern of faces in the dark environment is not rich enough, which leads to a large number of samples clustered around the manifold with smaller volumes, making it difficult to distinguish between faces. Our approach is not only to address the performance imbalance, but also to advance researchers' understanding of deep neural networks. Advances in science are usually accompanied by the establishment of new cognition.

(3) Our proposed semantic scale has great potential for application. In engineering applications, how many samples should be collected for each class is the most appropriate? When too few samples are collected, the class is under-represented, while too many will consume huge costs. Our approach can effectively solve this problem by stopping the collection when the semantic scales tend to be saturated. When we communicate with technology companies, we find that they have 100 million data, but there is no proper way to select representative data. So we design an idea to select representative data using semantic scales, the details of which are added in Appendix I.

## C EXPERIMENTS ON STANFORD POINT CLOUD MANIFOLDS

Since $(Z - Z_{mean})(Z - Z_{mean})^T$ is a real symmetric matrix, it is semi-positive definite. Further, $I + \frac{p}{m}(Z - Z_{mean})(Z - Z_{mean})^T$ is a positive definite matrix and therefore $det(I + \frac{p}{m}(Z - Z_{mean})(Z - Z_{mean})^T) > 0$. The semantic scale measure is derived from the singular value decomposition of the data matrix, which is jointly determined by most of the samples. Therefore, our method is insensitive to noisy samples, i.e., the semantic scale measure is numerically stable. The semantic scales of multiple Stanford point cloud manifolds with different sizes are calculated and plotted in Figure 6. Let the center point of bunny be $C_{bunny}$. We increase the volume of bunny by performing $w * (bunny - C_{bunny})$, and the other point clouds are scaled up in this manner. As the object manifold is scaled up, the calculated volume then increases slowly and monotonically, indicating that our method can accurately measure the relative size of the manifold volume and is numerically stable, an advantage that will help mitigate the effects of noisy samples.

## D EXPERIMENTAL DETAILS

### D.1 MARGINAL EFFECT OF SEMANTIC SCALE

We use the following method to generate new training datasets for classification experiments: assume that the total number of classes in the original dataset is $C$, and $m$ samples are randomly selected from each class to form a sub-dataset with a total number of samples of $C \times m$. The details of the sub-datasets generated based on CIFAR-10, CIFAR-100 and Mini-ImageNet are in Table 6.

### D.2 QUANTIFICATION OF SEMANTIC SCALE IMBALANCE

We train ResNet-18 and ResNet-34 on CIFAR-10-LT and CIFAR-10 with an imbalance factor of 200, respectively, and the test set of CIFAR-10-LT is consistent with CIFAR-10. During training, the batch size is fixed to 64, and the optimizer adopts Adam. The learning rate is initially 0.01 and becomes

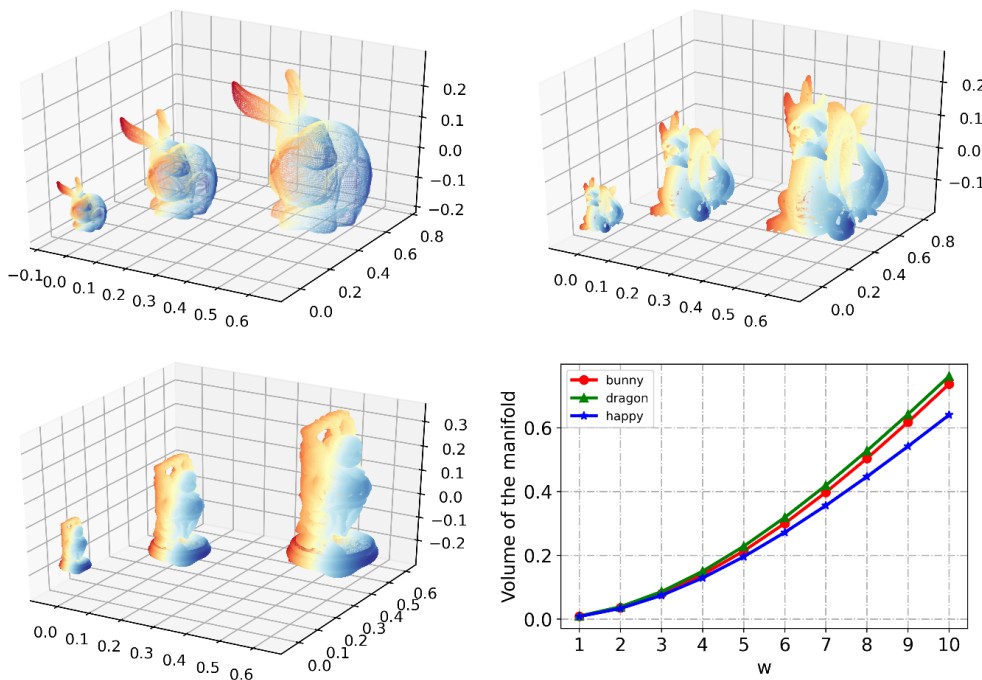

Figure 6: Increase three Stanford point cloud manifolds, and calculate their semantic scales.

Table 6: The sample-balanced sub-datasets with a total of 31. Among them, 13 sub-datasets are from CIFAR-10, 9 sub-datasets are from CIFAR-100, and the rest are from Mini-ImageNet. The test set remains the original test set. $C$ denotes the total number of classes in the original dataset and $m$ is the number of samples per class in the sub-dataset.

| Dataset | $C$ | Number | Sub-datasets | | | | | | | | | | | | |
|---|---|---|---|---|---|---|---|---|---|---|---|---|---|---|---|
| CIFAR-10 | 10 | $m$ | 50 | 100 | 200 | 300 | 400 | 500 | 600 | 800 | 1,000 | 2,000 | 3,000 | 4,000 | 5,000 |
| | | $C \times m$ | 500 | 1,000 | 2,000 | 3,000 | 4,000 | 5,000 | 6,000 | 8,000 | 10,000 | 20,000 | 30,000 | 40,000 | 50,000 |
| CIFAR-100 | 100 | $m$ | 10 | 20 | 30 | 50 | 100 | 200 | 300 | 400 | 500 | - | - | - | - |
| | | $C \times m$ | 1,000 | 2,000 | 3,000 | 5,000 | 10,000 | 20,000 | 30,000 | 40,000 | 50,000 | - | - | - | - |
| Mini-ImageNet | 100 | $m$ | 10 | 20 | 30 | 50 | 100 | 200 | 300 | 400 | 500 | - | - | - | - |
| | | $C \times m$ | 1,000 | 2,000 | 3,000 | 5,000 | 10,000 | 20,000 | 30,000 | 40,000 | 50,000 | - | - | - | - |

$0.98\times$the previous learning rate after each epoch. We do not employ other additional tricks and data augmentation strategies.

### D.3 SEMANTIC SCALE IMBALANCE ON LONG-TAILED DATA

We artificially produce two long-tailed versions of the MNIST dataset, called MNIST-LT-1 and MNIST-LT-2. The number of samples per class is listed in Table 7.

Figure 3 shows the class-wise accuracies of ResNet-18 and ResNet-34 trained on CIFAR-10-LT and CIFAR-100-LT with the same training settings as in Appendix D.2. Taking CIFAR-10 as an example, labels 1 to 10 correspond to: *airplane*, *automobile*, *bird*, *cat*, *deer*, *dog*, *frog*, *horse*, *ship*, *truck*. The prediction scores of classification experiments on CIFAR-10-LT and CIFAR-10 find that *cat* (label 4) is most easily confused with *dog* (label 6), as shown by their lowest accuracy on CIFAR-10 (Figure 4). However, the accuracy of *cat* and *dog* on CIFAR-10-LT is higher than that of *deer* (label 5), which is due to the dominant role of semantic scale $S'$ in $S$ for long-tailed data.

Table 7: The two long-tailed MNIST datasets resampled from MNIST.

| Dataset | MNIST-LT-1 | | | | | | | | | |
|---|---|---|---|---|---|---|---|---|---|---|
| Class label | 0 | 1 | 2 | 3 | 4 | 5 | 6 | 7 | 8 | 9 |
| Number | 5,923 | 3,590 | 2,940 | 2,518 | 2,256 | 1,972 | 1,700 | 1,300 | 1,100 | 900 |
| Dataset | MNIST-LT-2 | | | | | | | | | |
| Class label | 0 | 1 | 2 | 3 | 4 | 5 | 6 | 7 | 8 | 9 |
| Number | 5,923 | 3,090 | 2,540 | 1,818 | 1,356 | 972 | 484 | 272 | 122 | 74 |

### D.4 SEMANTIC SCALE IMBALANCE FOR MORE DATASETS

We have demonstrated the semantic scale imbalance on MNIST, MNIST-LT, CIFAR-10, CIFAR-10-LT, CIFAR-100 and CIFAR-100-LT in Section 3.4. Figure 7 additionally shows the degree of semantic scale imbalance on CUB-2011, Cars196, and Mini-ImageNet, indicating that the semantic scale imbalance is indeed prevalent in all kinds of datasets.

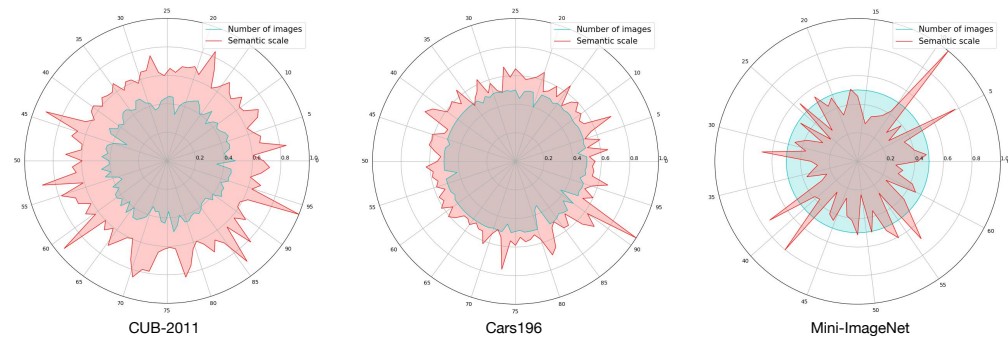

Figure 7: Semantic scale and number of samples per class. Different angles of the radar plot represent different classes, and the number of samples in the largest class is normalized to 0.5 for ease of observation.

### D.5 EXPERIMENTAL SETTINGS FOR SECTION 5.3 AND MORE EXPERIMENTS

#### D.5.1 EXPERIMENTAL SETTINGS FOR SECTION 5.3

**Backbone Network and Experimental Parameters**. Since we improve on the classical loss in the field of deep metric learning, we abide by the widely adopted backbone network, experimental parameters, and the division of datasets in this field. The BN-Inception [23; 53] pre-trained on ImageNet is adopted as the backbone network, and the training set is augmented by using random horizontal flipping and random cropping. All images are cropped to 224×224 as the input of the network. The output of the network after global average pooling is fed into a single fully connected layer to obtain 64- or 512-dimensional feature embeddings, and then all embeddings are clustered by K-means. The model is optimized by Adam with the batch size as 32 and the number of epochs as 50. We evaluate the performance of the learned embeddings with Recall@K and Normalized Mutual Information (NMI). The remaining experimental parameters used in the training are consistent with those reported in NormSoftmax and SoftTriple [42].

**Dataset Introduction (CUB-2011, Cars196 and CIFAR-100-LT)**. The CUB-2011 dataset has 5,864 images in the first 100 classes for training and 5,924 images in the second 100 classes for testing. The Cars196 dataset consists of 196 classes totaling 16,185 images, with the first 98 classes for training and the remaining classes for testing. The CIFAR-100 has 100 classes, each containing 600 images.

We create three long-tailed CIFAR-100 with the first 60 classes for training (See Figure 8a) and test on the remaining classes [42].

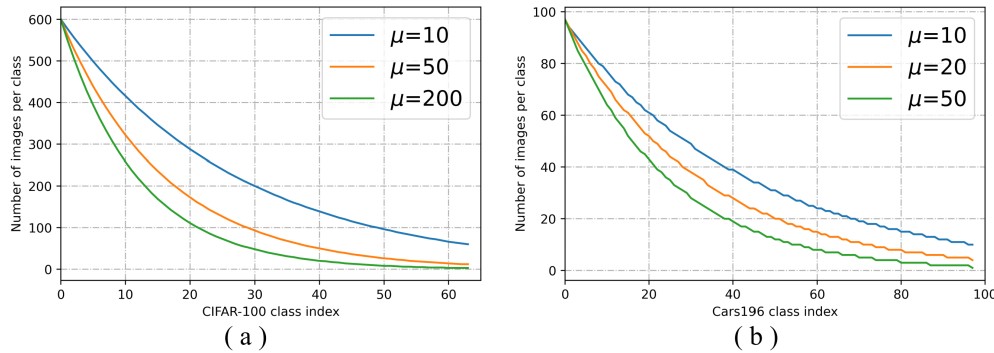

(a)                                                    (b)

Figure 8: Long-tailed CIFAR100 and Cars196 with different imbalance factors. We use the exponential function $n_i = N\mu^{i/(1-M)}$ to yield the number of training samples for each class, where $i$ is the class index (0-indexed), $N$ is the number of training samples in the largest class, $\mu$ is the imbalance factor, and $M$ is the total number of classes.

### D.5.2 MORE EXPERIMENTS

The long-tailed Cars196 is created using the first 98 classes for training (See Figure 8b) and the test set is the remaining classes. Both Mini-ImageNet and CIFAR-100 datasets contain 100 classes, each containing 600 samples. For the Mini-ImageNet dataset, the first 64 classes are the training set and the last 36 classes are the test set, and for the CIFAR-100 dataset, the first 60 classes are the training set and the remaining classes are the test set. Note that the experiments on CIFAR-100 in this section are different from the classification experiments on CIFAR-100 in Sec 5.3. The purpose of this experiments is to complement the effectiveness of our proposed method on both long-tailed and sample-balanced datasets for the field of deep metric learning.

Table 8: Comparison on long-tailed Cars196.

| Dataset | | Long-tailed Cars196 | | | | | | | | |
|---|---|---|---|---|---|---|---|---|---|---|
| Imbalance factor | | | 10 | | | 20 | | | 50 | |
| Metric | | R@1 | R@2 | NMI | R@1 | R@2 | NMI | R@1 | R@2 | NMI |
| dim 64 | NormSoftmax | 66.4 | 76.9 | 58.9 | 63.1 | 74.2 | 54.9 | 59.5 | 71.1 | 52.9 |
| | CB-NSM | 68.9 | 78.0 | 60.1 | 64.9 | 75.2 | 56.1 | 61.7 | 72.5 | 53.3 |
| | **DSB-NSM** | **69.5** | **78.6** | **60.7** | **65.4** | **75.7** | **56.6** | **62.3** | **73.0** | **54.1** |
| | SoftTriple | 70.2 | 80.5 | 61.4 | 64.7 | 75.8 | 57.5 | 62.9 | 74.1 | 55.2 |
| | CB-ST | 71.9 | 81.3 | 62.9 | 66.5 | 76.9 | 58.6 | 64.8 | 75.4 | 56.0 |
| | **DSB-ST** | **72.3** | **81.8** | **63.4** | **66.8** | **77.5** | **59.7** | **65.4** | **75.3** | **56.6** |

Table 8 shows the performance comparison of DSB-NSM and DSB-ST on the long-tailed Car196. When the imbalance factor is 10, DSB-NSM outperforms NSM (NormSoftmax) by **3.1%** on R@1 and DSB-ST outperforms ST (SoftTriple) by **2.1%**. When the imbalance factor is 50, DSB-NSM and DSB-ST improve **2.8%** and **2.5%**, respectively, on R@1 compared to the original method. In addition, DSB loss performs better than CB loss on all metrics.

Table 9 shows that compared with the original losses, DSB-NSM and DSB-ST are able to consistently improve R@1 by **1.3%** on average and NMI by **1-2%** for both sample-balanced datasets, with all other metrics outperforming the original. The results in Tables 8 and 9 further confirm that our proposed dynamic semantic-scale-balanced learning is applicable to long-tailed and sample-balanced datasets in the field of deep metric learning, and has broad application prospects.

Table 9: Comparison on Mini-ImageNet and CIFAR-100.

| Dataset | Mini-ImageNet | | | CIFAR-100 | | |
|---|---|---|---|---|---|---|
| Metric | R@1 | R@2 | NMI | R@1 | R@2 | NMI |
| NormSoftmax | 85.7 | 91.2 | 74.1 | 60.1 | 71.5 | 49.4 |
| **DSB-NSM** | **87.1**(+1.4) | **92.0**(+0.8) | **75.5**(+1.4) | **61.4**(+1.3) | **72.2**(+0.7) | **50.6**(+1.2) |
| SoftTriple | 86.9 | 92.0 | 77.3 | 62.1 | 73.3 | 52.0 |
| **DSB-ST** | **88.0**(+1.1) | **92.8**(+0.8) | **78.8**(+1.5) | **63.5**(+1.4) | **73.9**(+0.6) | **53.0**(+1.0). |

## D.6 RESULTS ON THE FUNDUS DATASETS OIA-ODIR AND OIA-ODIR-B

### D.6.1 DATASET INTRODUCTION

The OIA-ODIR dataset [31] was made public in 2019, and it contains a total of 10,000 fundus images in 8 classes. As shown in Figure 9, the eight classes are: Normal(N), hypertensive retinopathy(D), glaucoma(G), cataract(C), agerelated macular degeneration(A) , hypertension complication (H), pathologic myopia (M), other disease / abnormality(O). Considering that O usually appears together with other diseases, to reduce ambiguity, we adopt the data splitting scheme of [71], using only the data of the first 7 classes, and the number of training samples and test samples for each class is shown in Figure 10.

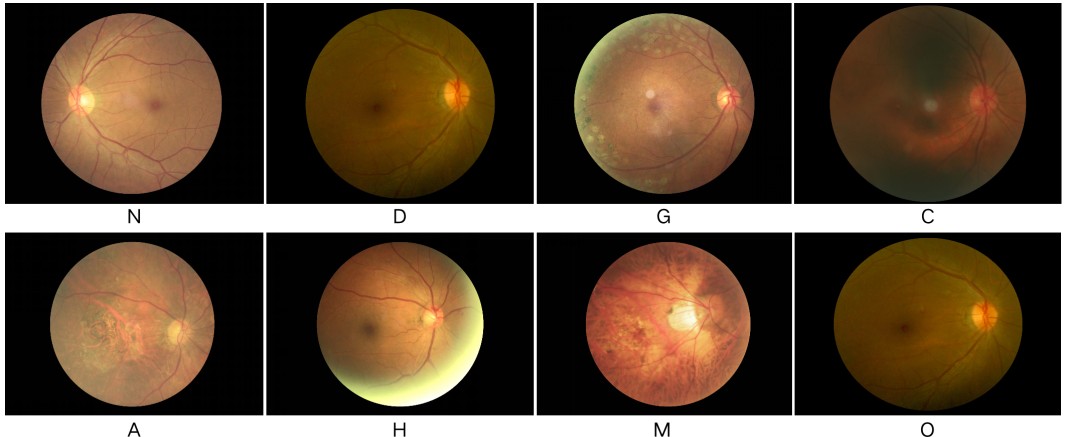

Figure 9: Eight fundus images in the OIA-ODIR dataset.

The OIA-ODIR dataset suffers from an unbalanced number of samples. To fully validate our method, we produced a balanced version of the OIA-ODIR dataset, OIA-ODIR-B, by using the class with the least number of samples as the benchmark. As shown in Figure 10, each class of OIA-ODIR-B contains 103 training samples and 46 test samples.

In addition to the number of samples, we plot the degree of semantic scale imbalance for the training sets of OIA-ODIR and OIA-ODIR-BS in Figure 10.

### D.6.2 BACKBONE NETWORK AND EXPERIMENTAL PARAMETERS

We used ResNet-50, pre-trained on ImageNet, as the backbone network. An adam optimizer with a learning rate of 0.1 (linear decay), a momentum of 0.9, and a weight decay factor of 0.005 was adopted to train all networks. In keeping with [71], average precision (AP) was used as the performance metric of the model.

### D.6.3 RESULTS ON OIA-ODIR

We improved the advanced class rebalancing method (BS [43], Focal loss [11], LDAM [4]) and the classification results are plotted in Figure 11. The experimental findings are summarized as follows.

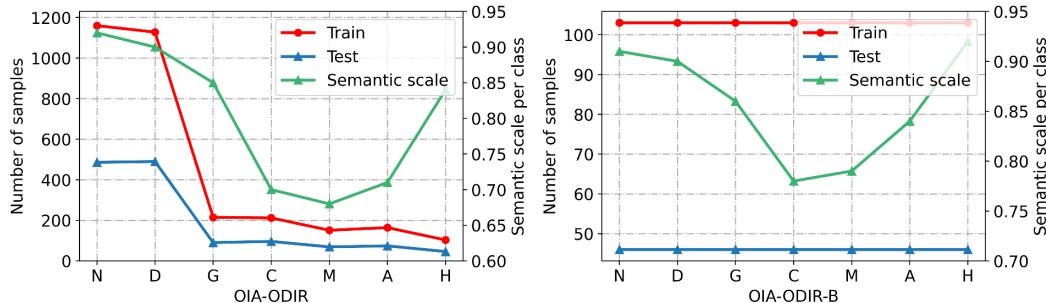

Figure 10: The number of training and test samples for each category and the degree of semantic scale imbalance in OIA-ODIR and OIA-ODIR-B.

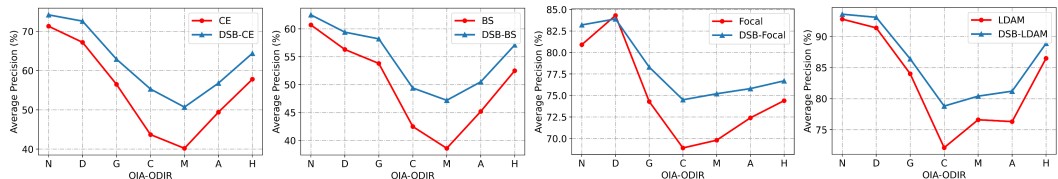

Figure 11: The enhancement effect of our method for CE, BS, Focal, and LDAM on the OIA-ODIR.

- Although the sample from class H is the smallest, all methods outperform on class H than on class C, class M, and class A. This again shows that the number of samples is not the best measure of class imbalance.

- Methods based on sample numbers usually result in larger boosts for the classes with the smallest sample numbers and thus fail to give more attention to class C, class M, and class A. Our method has the most significant boosts for these three classes, indicating that semantic scale imbalance can more accurately reflect the difficulty of the classes.

### D.6.4 RESULTS ON OIA-ODIR-B

Since the class rebalancing method based on the number of samples cannot be applied to the dataset with a balanced number of samples, we additionally adopted VGG-16, ResNet-18 and SE-ResNet-50 as the backbone network to test the effect of DSB-CE on CE enhancement, and the experimental results are shown in Figure 12. The experimental findings are summarized as follows.

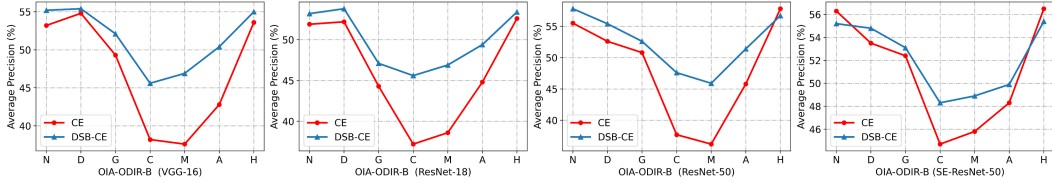

Figure 12: Performance gains from our approach for multiple backbone networks on OIA-ODIR.

- With a balanced number of samples, the model still performs poorly on class C, class M and class A. Figure 10 shows that the semantic scales of these three classes are significantly smaller than the other classes.

- Our approach results in significant performance gains for all models on class C, class M, and class A, and promotes more balanced model performance on all classes, which is important in medical AI.

**Experiment Summary.** We validated the effectiveness of semantic-scale-balanced learning both on a dataset of fundus images with balanced sample numbers and on a long-tailed dataset of fundus images. Experimental results show that semantic scale imbalance exists in medical image datasets

and significantly limits the performance of deep neural networks, so it is necessary to introduce semantic-scale-balanced learning in medical image classification.

## D.7 REMOTE SENSING IMAGE SCENE CLASSIFICATION

In this section, we validate the effectiveness of semantic-scale-balanced learning in a sample-balanced remote sensing image classification task, which demonstrates the necessity of introducing semantic scale imbalance into the field of remote sensing image recognition.

### D.7.1 DATASET INTRODUCTION

- **RSSCN7** dataset contains $2,800$ remote sensing images which are classified into 7 typical scene categories: grassland, forest, farmland, parking lot, residential region, industrial region, and river and lake. Figure 13 shows the images of the seven scenarios. Following the official split, the number of images for training and testing is 50% of the total number each.

- **NWPU-RESISC45** dataset contains a total of $31,500$ images with pixel size of $256 \times 256$, covering 45 scene classes with 700 images in each class. This dataset has large intra-class variability and inter-class similarity due to the large differences in image spatial resolution, untitled pose, and illumination. Following the official split, 20% of the images are used for training and 80% for testing.

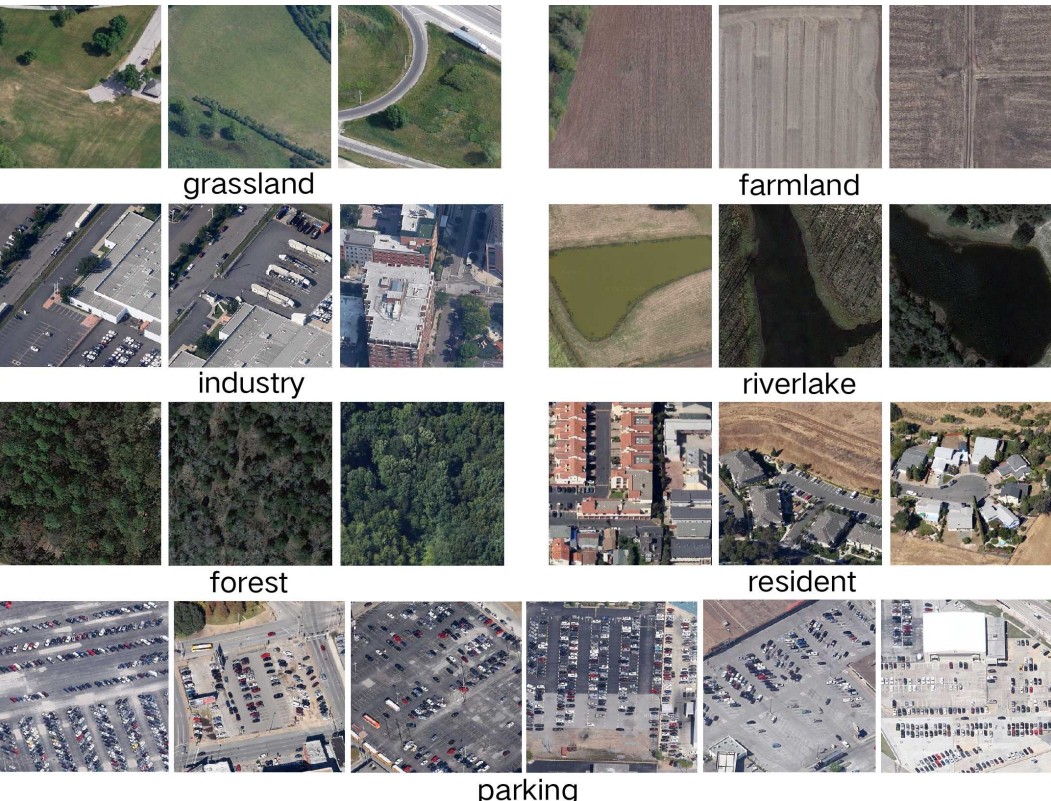

Figure 13: The seven scenarios are included in the RSSCN7 dataset.

### D.7.2 BACKBONE NETWORK AND EXPERIMENTAL PARAMETERS

We select VGG-16, GoogLeNet, and ResNet-34 as the backbone networks. The Adam optimizer (default parameter) is adopted to update the model until convergence, and the learning rate decays 10 times every 50 epochs. In addition, the batch size is set to 100 and no data augmentation is used throughout the training process.

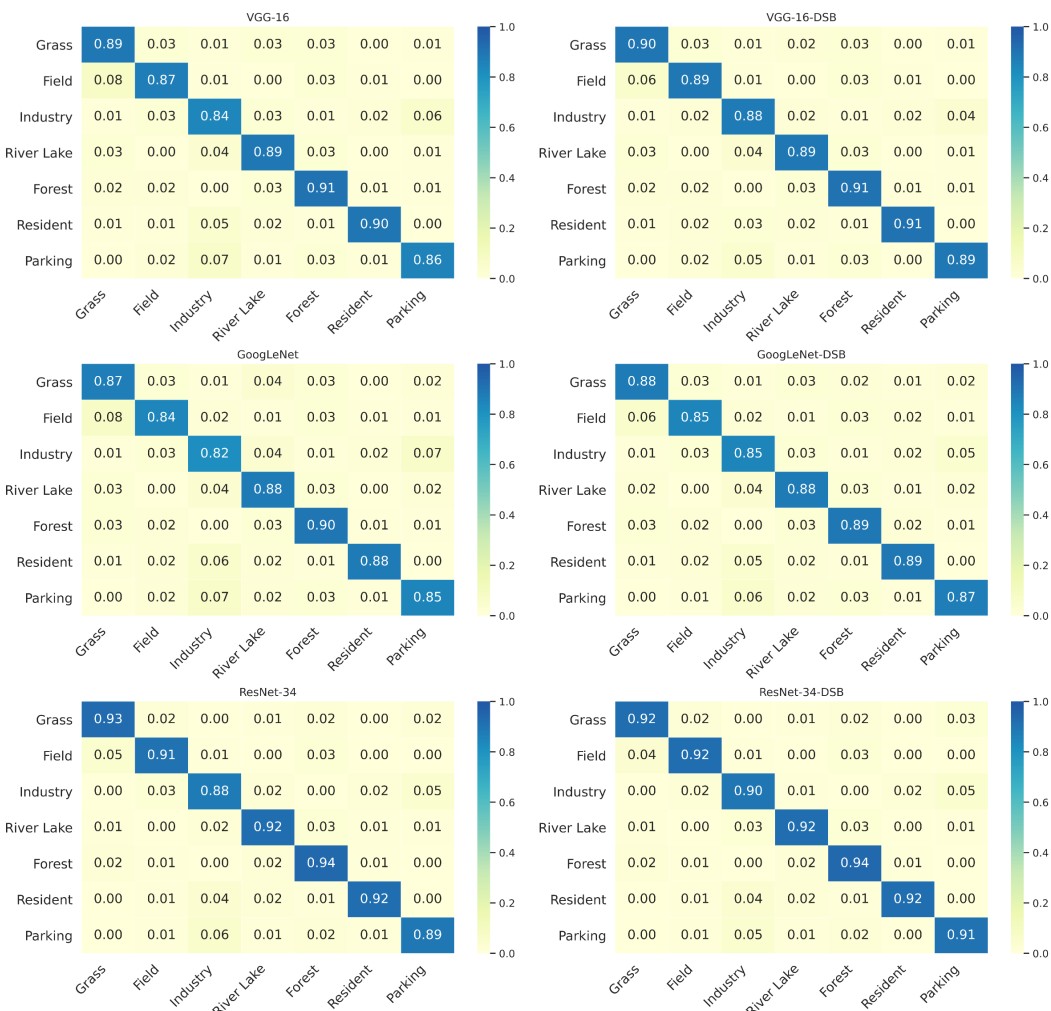

Figure 14: **Left column:** confusion matrix of VGG-16, GoogLeNet, and ResNet-34 on the RSSCN7 dataset. **Right column:** confusion matrix of VGG-16-DSB, GoogLeNet-DSB, and ResNet-34-DSB on the RSSCN7 dataset.

### D.7.3 RESULTS ON RSSCN7

We trained all backbone networks with dynamic semantic-scale-balanced learning. The experimental results are shown in Figure 14. It can be found that our method improves the performance of all models. When our method is not employed, all backbone networks are significantly weaker in recognizing industrial regions than other scenes. Our method makes the recognition ability of the model for different scenes more balanced, thus improving the overall performance of the model.

Specifically, dynamic semantic-scale-balanced learning improves VGG-16's recognition accuracy for industrial regions and parking lots by $4\%$ and $3\%$, respectively, significantly reducing the bias of the model. Dynamic semantic-scale-balanced learning also performs well on GoogLeNet and ResNet-34, where it improves the overall accuracy of GoogLe and ResNet by $1\%$ and $0.6\%$, respectively.

### D.7.4 RESULTS ON NWPU-RESISC45

We significantly improved the performance of multiple backbone networks by employing dynamic semantic-scale-balanced learning on the NWPU-RESISC45 dataset, and the experimental results are illustrated in Figure 15. It can be observed that the overall performance of VGG-16-DSB is $1.8\%$ higher than that of VGG-16. Meanwhile, dynamic semantic-scale-balanced learning improves the overall performance of GoogLeNet and ResNet-34 by $1.6\%$ and $1.2\%$.

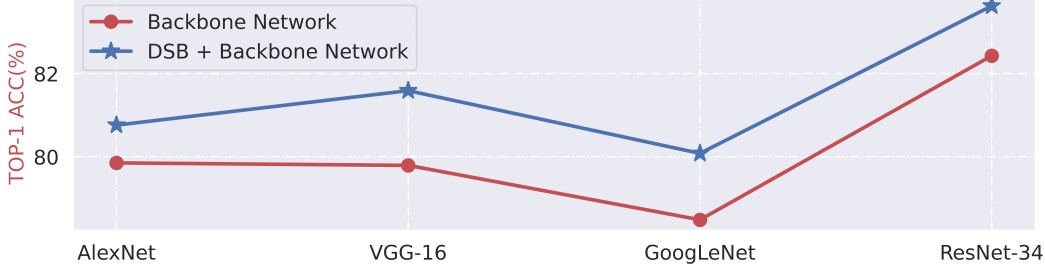

Figure 15: Comparison of four backbone networks before and after combining with dynamic semantic scale-balanced learning on dataset NWPU-RESISC45.

**Experiment Summary.** On two remote sensing image datasets with balanced sample numbers, our method shows significant improvements on common backbone networks. The experimental results show that semantic scale imbalance exists in the remote sensing image dataset and affects the performance of deep neural networks to some extent. Remote sensing images hold great promise for applications in agriculture, industry, and the military, so it is crucial to promote the fairness of deep neural networks on remote sensing images.

# E  PSEUDO CODE FOR SAMPLE VOLUME

An image can be considered as a point in the sample space, and the dimension of the sample space is the same as the number of image pixel points. The manifold distribution law considers that multiple images from a class are distributed around a low-dimensional manifold in the sample space. We calculate for each class the volume of its corresponding manifold and call it the sample volume. We provide the pseudo code for the calculation of the sample volume in Algorithm 1. In this work, we resize the image to (16, 16, 3) and then calculate the sample volume after flattening.

---

**Algorithm 1** Calculation of Sample Volume

---

**Input:** Training set $D = \{(x_i, y_i)\}_{i=1}^{M}$ with the total number $C$ of classes
**Onput:** Sample volumes for all classes
**for** $j = 1$ to $C$ **do**
  Select the sample set $D_j = \{(x_i, y_i)\}_{i=1}^{m_j}$ for class $j$ from $D$, $m_j$ is the number of samples for class $j$
  Resize the image to $(imagesize, imagesize, 3)$
  Flatten the image into a vector of length $d = imagesize \times imagesize \times 3$ and store it in $Z_j = [z_1, z_2, \ldots, z_{m_j}] \in \mathbb{R}^{d \times m_j}$
  $Z_j = Z_j - NumPy.mean(Z_j, 1)$
  Calculate the covariance matrix $\Sigma_j = \frac{1}{m_j} Z_j Z_j^T$
  Calculate the sample volume $Vol(\Sigma_j) = \frac{1}{2} \log_2 \det(I + d\Sigma_j)$ for class $j$
**end for**

---

# F  DYNAMIC SEMANTIC-SCALE-BALANCED LEARNING

## F.1  DSB-NSM, DSB-ST AND DSB-FOCAL LOSS

Given the embedding $z$ of a sample and label $y_i$, the dynamic semantic-scale-balanced (DSB) loss can be expressed as:

$$DSB(z, y_i) = \frac{1}{S_i} L(z, y_i), i = 1, 2, \ldots, C, \tag{4}$$

where $y_i$ is the label of the sample from class $i$. To show how to combine the general loss to generate the dynamic semantic-scale-balanced loss, we improve the NormSoftmax (NSM) cross-entropy loss and SoftTriple (ST) loss. NormSoftmax removes the bias term in the last linear layer and an L2 normalization module is added to the inputs and weights before the SoftMax loss.

$[w_1, w_2, \cdots, w_C] \in \mathbb{R}^{d \times C}$ is the last fully connected layer, then the DSB-NSM with temperature $\sigma$ generated by embedding $z$ can be written as:

$$DSB-NSM(z, y_i) = -\frac{1}{S_i} \log\left(\frac{\exp(w_{y_i}^T z/\sigma)}{\sum_{j=1}^{C} \exp(w_j^T z/\sigma)}\right). \tag{5}$$

The SoftTriple loss combined with the semantic-scale-balanced term is expressed as

$$DSB-ST(z, y_i) = -\frac{1}{S_i} \log\left(\frac{\exp(\lambda(D'_{z,y_i} - \delta))}{\exp(\lambda(D'_{z,y_i} - \delta)) + \sum_{j \neq y} \exp(\lambda D'_{i,j})}\right), \tag{6}$$

where $\lambda$ is a scaling factor and $\delta$ is a hyperparameter. The relaxed similarity between embedding $z$ and class $c$ is defined as $D'_{z,c} = \sum_k \frac{\exp(\frac{1}{\gamma} z^T w_c^k)}{\sum_k \exp(\frac{1}{\gamma} z^T w_c^k)} z^T w_c^k$, where $k$ is the number of centers for each class.

The purpose of Focal loss is to apply small loss weights to samples with high classification confidence, thus increasing the proportion of loss of hard samples with low classification confidence to the overall loss. The $\alpha$-balanced variant of Focal loss regulates the proportion of loss among samples while assigning different weights to each class, which is denoted as $FL(p_t) = -\alpha_t(1 - p_t)^\gamma \log(p_t)$, where $p_t$ is the probability that the sample belongs to the true class. When $\alpha_t = \frac{1}{S_i}$, Focal loss is transformed into DSB-Focal loss.

### F.2 Dynamic Re-Weighting Training Framework

Given the training samples $X = [x_1, x_2, \ldots, x_N]$ containing $C$ classes and corresponding labels $Y = [y_1, y_2, \ldots, y_N]$, the number of samples per class is $N_i$ $(i = 1, 2 \ldots, C)$, and the total number of samples is $N$. The d-dimensional features extracted by the CNNs are denoted as $Z = [z_1, z_2, \ldots, z_N] \in \mathbb{R}^{d \times N}$. In this work, we conduct experiments for two types of tasks, image classification and deep metric learning. In the field of deep metric learning, 64-dimensional features are generally adopted, while the features extracted by the network in image classification tasks tend to be of high dimensionality. For example, the feature dimension extracted by ResNet-50 is 2048, which will occupy more video memory. Therefore, when saving historical features in the classification task, one-dimensional average pooling is performed on all features to reduce the feature dimension to 64, which is consistent with the common feature dimension in the field of deep metric learning while preserving the geometry of the distribution (because the pooling operation is translation invariant, rotation invariant, and scale invariant).

In the following, we describe the three-stage training framework in detail.

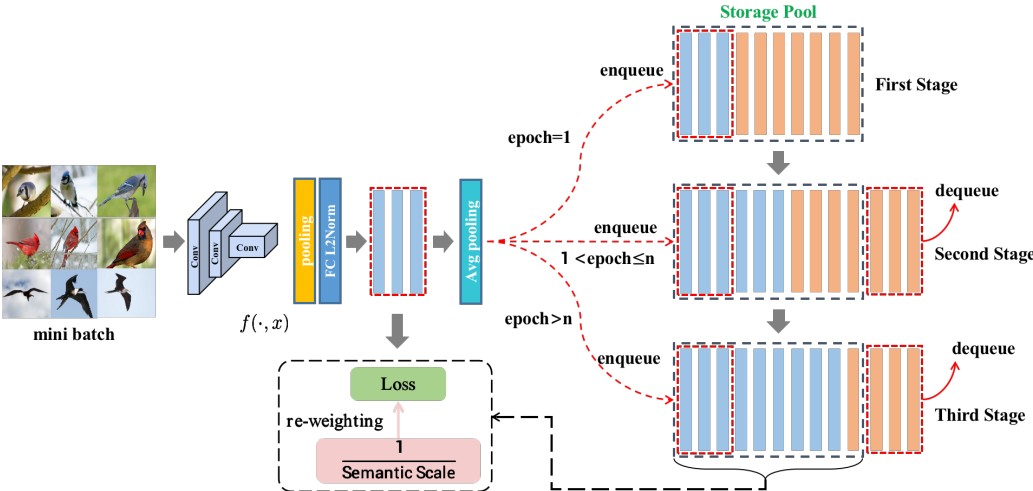

Figure 16: The three-stage training framework. The features in the storage pool are continuously updated during training and semantic scales are calculated using all the latest features.

The three-stage training framework is shown in Figure 16:

**(1)** In the **first stage**, all the features and labels generated by the 1st epoch are stored in $Q$, which is denoted as:

$$Q = \begin{bmatrix} Z \\ Y \end{bmatrix} = \begin{bmatrix} Z_{11} & \cdots & Z_{1N} \\ \vdots & \ddots & \vdots \\ Z_{d1} & \cdots & Z_{dN} \\ y_1 & \cdots & y_N \end{bmatrix} \in \mathbb{R}^{(d+1) \times N}.$$

$Q$ contains the features and labels of all samples, but in the early stage of training, the historical features have a large drift from the current features and cannot be used directly to calculate the semantic scale.

**(2)** The **second stage** corresponds to epoch 2 to epoch $n$. At each iteration, the oldest mini-batch features and labels in $Q$ are removed and those generated by the current iteration are stored. The goal is to continuously update the features in $Q$ until the feature drift is small enough. We set $n$ to 5 in our experiments, and the original loss function is used in the first two stages. Figure 5 shows the effect of $n$ on the model performance. A larger $n$ does not hurt the model performance, but only takes a little more time. Experience suggests that setting $n$ to 5 is sufficient.

**(3)** The **third stage** corresponds to epoch $> n$. At each iteration, the semantic scales are calculated using the features in $Q$ after updating $Q$, and the original loss is re-weighted.

Algorithm 2 shows how to apply the dynamic re-weighting training framework by taking DSB-ST loss as an example.

---

**Algorithm 2** Dynamic Re-Weighting Training Framework

---

**Input:** Training set $D = \{(x_i, y_i)\}_{i=1}^{M}$, total training epochs $N_{epoch}$, defined encoder is model()
Initialize the queue $Q$
**for** $epoch = 1$ to $N_{epoch}$ **do**
  **for** $iteration = 0$ to $\frac{M}{batchsize}$ **do**
    Sample a mini-batch $\{(x_i, y_i)\}_{i=1}^{batchsize}$ from $D$
    Calculate features $F = [f_1, \ldots, f_i, \ldots, f_{batchsize}], f_i = \text{model}(x_i), i = 1, \ldots, batchsize$
    Store $F$ and label $y$ into $Q$: $enqueue(Q, [F, y])$
    **if** $epoch < n$ **then**
      **if** $epoch > 1$ **then**
        Dequeue the oldest mini-batch features from $Q$
      **end if**
      Calculate loss $L = SoftTripleloss(F, y)$
    **else**
      Dequeue the oldest mini-batch features from $Q$
      Calculate loss $L = DSB.SoftTripleloss(F, y)$
    **end if**
    Perform back propagation: $L$.backward()
    optimizer.step()
  **end for**
**end for**

---

The proposed three-stage training framework overcomes the difficulty of not being able to calculate class-wise semantic scales during training due to the limited number of samples per batch. In fact, there is a simple and brute-force method to achieve the goal of calculating class-wise semantic scales in real time, which is to extract features of all samples using the current model after each iteration. However, this would take a lot of time, for example, when training ImageNet with batch size of 512, one epoch contains about 2,500 iterations. This also means that all features need to be extracted 2,500 times, which is unacceptable.

Our method causes almost no reduction in training speed. In terms of video memory consumption, even the extracted features of a million-level dataset like ImageNet can all be placed on one graphics card (about 6,000 MB of video memory is needed). In this work, we use 4 NVIDIA 2080Ti GPUs to train all the models. A comparison of the video memory consumption and training speed for some

Table 10: Comparison of DSB-ST and SoftTriple in terms of memory consumption and training speed. The speed is measured by the average number of iterations per second. The additional video memory consumption due to our method is almost negligible.

| Dataset | GPU Memory | | Training speed | |
|---|---|---|---|---|
| | SoftTriple | DSB-ST | SoftTriple | DSB-ST |
| ImageNet-LT | 24.29 GB | 24.75 GB | 6.01 it/s | 5.56 it/s |
| iNaturalist2018 | 45.13 GB | 46.88 GB | 3.32 it/s | 3.03 it/s |
| Cars196 | 3491 MB | 4097 MB | 20.21 it/s | 18.95 it/s |
| CUB-2011 | 3225 MB | 3647 MB | 21.95 it/s | 18.58 it/s |
| Mini-ImageNet | 3491 MB | 3713 MB | 23.34 it/s | 20.36 it/s |

experiments is shown in Table 10. It can be noticed that the consumption of our method is negligible for the video memory, and the training speed is about 90% of the original method.

## G  Volume Formula for the Low-Dimensional Parallel Hexahedron in the High-Dimensional Space

In Section 3.2 of the main content, we deduce from the singular value decomposition of the matrix $Z = [z_1, z_2, \ldots, z_m] \in \mathbb{R}^{d \times m}$ composed of features that the volume $Vol(Z)$ of the subspace spanned by $z_i$ is proportional to $\sqrt{\det(\frac{1}{m} ZZ^T)}$. Here, we assume that in $\mathbb{R}^d$, given $m$ d-dimensional vectors, these vectors will define a parallel hexahedron in $\mathbb{R}^n$. The problem is how to calculate the parallel hexahedron. For example, consider two vectors

$$z_1 = \begin{bmatrix} 1 \\ 2 \\ 3 \end{bmatrix}, z_2 = \begin{bmatrix} 3 \\ 2 \\ 1 \end{bmatrix}.$$

The parallel hexahedron defined by these two vectors is a parallelogram in $\mathbb{R}^3$. We want to find a formula to calculate the area of the parallelogram. (Note that the true three-dimensional volume of the planar parallelogram is 0, just as the length of the point is 0 and the area of the line is 0. Here, we are trying to measure the two-dimensional "volume" of the parallelogram.)

Next we will introduce two special cases of parallel hexahedral volume, for a single vector

$$z = \begin{bmatrix} a_1 \\ \vdots \\ a_n \end{bmatrix} \in \mathbb{R}^n,$$

whose parallel hexahedral is itself. Here "volume" means the length of the vector, and according to the Pythagorean theorem its volume is

$$\sqrt{a_1^2 + \cdots + a_n^2}. \tag{7}$$

Another case is to give $n$ vectors in $\mathbb{R}^n$. Suppose that these $n$ vectors are

$$z_1 = \begin{bmatrix} a_{11} \\ \vdots \\ a_{n1} \end{bmatrix}, \ldots, z_n = \begin{bmatrix} a_{1n} \\ \vdots \\ a_{nn} \end{bmatrix},$$

we can know that the volume of the resulting parallel hexahedron is

$$\left| \det \begin{bmatrix} a_{11} & \cdots & a_{1n} \\ \vdots & \ddots & \vdots \\ a_{n1} & \cdots & a_{nn} \end{bmatrix} \right|. \tag{8}$$

In the non-special case, the formula for the volume of a low-dimensional parallel hexahedron in a high-dimensional space will contain Results (7) and (8). Here, we first present the final formula and then discuss why it is reasonable. Write the $k$ vectors $z_1, \ldots, z_k$ in $\mathbb{R}^n$ as column vectors. Let

$$Z = [z_1, \ldots, z_k] \in \mathbb{R}^{n \times k},$$

and the volume of the parallel hexahedron derived from the vectors $z_1, \ldots, z_k$ is

$$\sqrt{\det[Z^T Z]}.$$

We now discuss why $\sqrt{\det[Z^T Z]}$ must be the volume in the general case.

**Lemma 1** For a matrix $Z = [z_1, \ldots, z_k]$, we can get

$$Z^T Z = \begin{bmatrix} |z_1|^2 & z_1 \cdot z_2 & \cdots & z_1 \cdot z_k \\ \vdots & \vdots & \ddots & \vdots \\ z_k \cdot z_1 & z_k \cdot z_2 & \cdots & |z_k|^2 \end{bmatrix},$$

where $z_i \cdot z_j$ denotes the dot product of the vectors $z_i$ and $z_j$ and $|z_i| = \sqrt{z_i \cdot z_i}$ denotes the length of the vector.

The proof of Lemma 1 needs to focus only on

$$Z^T Z = \begin{bmatrix} z_1^T \\ \vdots \\ z_k^T \end{bmatrix} [z_1, \cdots, z_k].$$

If we apply any linear transformation that preserves angularity and length in $\mathbb{R}^n$ (in other words, if we perform a rotation operation on $\mathbb{R}^n$), the numbers $|z_i|$ and $z_i \cdot z_j$ do not change. The multiple sets of all linear transformations that preserve angle and length in $\mathbb{R}^n$ form a group, called the orthogonal group and denoted as $\mathrm{O}(n)$. This allows us to reduce the problem to that of finding the volume of a parallel hexahedron in $\mathbb{R}^k$.

***Proof.*** It is known that

$$\sqrt{\det[Z^T Z]} = \sqrt{\det \begin{bmatrix} |z_1|^2 & z_1 \cdot z_2 & \cdots & z_1 \cdot z_k \\ \vdots & \vdots & \ddots & \vdots \\ z_k \cdot z_1 & z_k \cdot z_2 & \cdots & |z_k|^2 \end{bmatrix}}.$$

To prove that the above equation must be the formula for the volume, we first consider a set of standard basis of $\mathbb{R}^n$:

$$e_1 = \begin{bmatrix} 1 \\ 0 \\ \vdots \\ 0 \end{bmatrix}, e_2 = \begin{bmatrix} 0 \\ 1 \\ \vdots \\ 0 \end{bmatrix}, \ldots, e_n = \begin{bmatrix} 0 \\ 0 \\ \vdots \\ 1 \end{bmatrix}.$$

According to Lemma 1, we are able to find a rotation of $\mathbb{R}^n$, which is able to maintain both length and angle, and also rotate our vectors $z_1, \ldots, z_k$ such that they can be fully represented linearly by the first $k$ standard vectors $e_1, \ldots, e_k$ (which is geometrically reasonable). After the rotation, the latter $n-k$ dimensions of each vector zi are 0. Therefore we can think of our parallel hexahedron as consisting of $k$ vectors in $\mathbb{R}^k$ and we already know how to calculate it, which is

$$\sqrt{\det \begin{bmatrix} |z_1|^2 & z_1 \cdot z_2 & \cdots & z_1 \cdot z_k \\ \vdots & \vdots & \ddots & \vdots \\ z_k \cdot z_1 & z_k \cdot z_2 & \cdots & |z_k|^2 \end{bmatrix}}.$$

# H MORE ANALYSIS

In this section, we will add the experimental results of the following three questions.

(1) The effectiveness of dynamic semantic-scale-balanced learning without considering inter-class interference.

(2) Comparison with other methods of measuring class-level difficulty.

(3) Dividing ImageNet into three subsets based on semantic scale, and showing the performance of dynamic semantic-scale-balanced learning on the three subsets.

## H.1 THE EFFECTIVENESS OF DYNAMIC SEMANTIC-SCALE-BALANCED LEARNING WITHOUT CONSIDERING INTER-CLASS INTERFERENCE

We have weakened the effect of inter-class interference when designing the measurement of semantic scale imbalance. The semantic scale of $m$ classes after maximum normalization is assumed to be $S' = [S'_1, S'_2, \ldots, S'_m]^T$, and the centers of all classes are $O = [o_1, o_2, \ldots, o_m]^T$. Define the distance between the centers of class $i$ and class $j$ as $d_{i,j} = \|o_i - o_j\|_2$, the weight $w_i = \frac{1}{m-1} \sum_{j=1}^{m} \|o_i - o_j\|_2$. The weights of $m$ classes are written as $W' = [w_1, w_2, \ldots, w_m]^T$. After the maximum normalization and logarithmic transformation of $W'$, we can obtain $W = \log_2 (\alpha + W')$, $\alpha \geq 1$, where $\alpha$ is used to control the smoothing degree of $W$. After considering the inter-class distance, the semantic scale $S = S' \odot W$, and the role of $S'$ in dominating the degree of imbalance is greater when $\alpha$ is larger. The second-order derivative of the function $W = \log_2 (\alpha + W')$ is less than 0, so the increment of $W$ decreases as $\alpha$ increases. When the value of $\alpha$ is taken to be large, $W'$ hardly works.

The Pearson correlation coefficients between class accuracy and inter-class interference, semantic scale, and semantic scale considering inter-class interference, respectively, are shown in Table 1. It can be seen that the Pearson correlation coefficient between semantic scale and class accuracy on CIFAR-10-LT without considering inter-class interference still reaches $0.8688$, while the correlation coefficient between inter-class distance $W$ and class accuracy is only $0.2957$, which illustrates the importance of semantic scale. In addition, we have added the correlation coefficients between the effective sample numbers and class accuracies in Table 1. It can be observed that the correlation between effective sample number and class accuracy is almost the same as the correlation between sample number and class accuracy, which is due to the fact that effective sample number is a monotonic function of sample number. To demonstrate the performance of dynamic semantic-scale-balanced learning without considering inter-class interference in more detail, we conducted experiments on ImageNet-LT. The experimental settings are the same as those in Table 2.

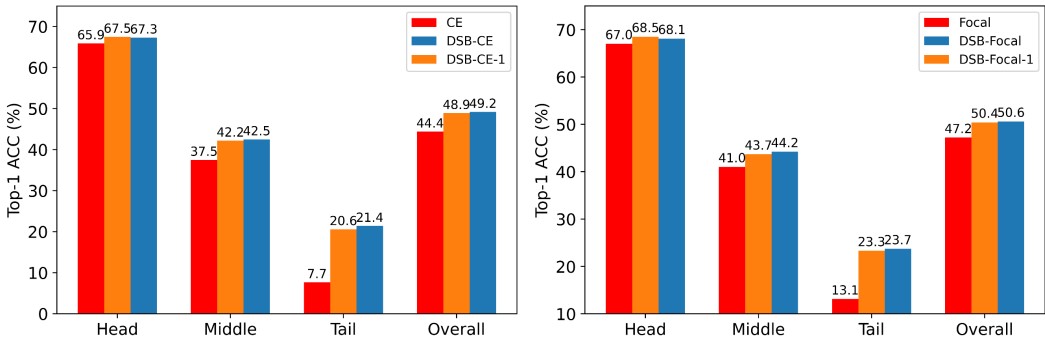

Figure 17: Accuracy comparison on ImageNet-LT.

The dynamic semantic-scale-balanced loss without considering inter-class interference is denoted as DSB-CE-1 and DSB-Focal-1. The experimental results are shown in Figure 17. It can be observed that DSB-CE-1 and DSB-Focal-1 have almost no performance degradation compared to DSB-CE and DSB-Focal. The above observation is as expected, since our recent study shows that the correlation between the separation degree of feature manifolds and the accuracy of the corresponding class decreases during training and that the existing model can eliminate the main effect of separation degree between feature manifolds on model bias.

H.2    Comparison with other methods of measuring class-level difficulty

Difficult example mining [47; 58] is an instance-level approach, while we focus on class-level difficulty. We note that a recent work on measuring class difficulty (CDB loss [37]) was published in IJCV, which can be compared with our work. In addition, LOCE [14] and domain balancing [2] also measure class-level difficulty, but LOCE is designed for object detection tasks, so we compare semantic-scale-balanced learning with CDB loss and domain balancing. The description of CDB loss, LOCE and domain balancing is as follows.

- The imbalance in class performance is referred to as the "bias" of the model, and [37] defines the model bias as
$$bias = \max(\frac{\max_{c=1}^{N} A_c}{\min_{c'=1}^{N} A_{c'} + \varepsilon} - 1, 0),$$
where $A_c$ denotes the accuracy of the $c$-th class. When the accuracy of each class is identical, bias = 0. [37] computes the difficulty of class c using $1 - A_c$ and calculates the weights of the loss function using a nonlinear function of class difficulty.

- LOCE [14] uses the mean classification prediction score to monitor the learning status for different classes and apply it to guide class-level margin adjustment for enhancing tail-class performance [78].

- Domain balancing [2] studied a long-tailed domain problem, where a small number of domains (containing multiple classes) frequently appear while other domains exist less. To address this task, this work introduced a novel domain frequency indicator based on the inter-class compactness of features, and uses this indicator to re-margin the feature space of tail domains [78].

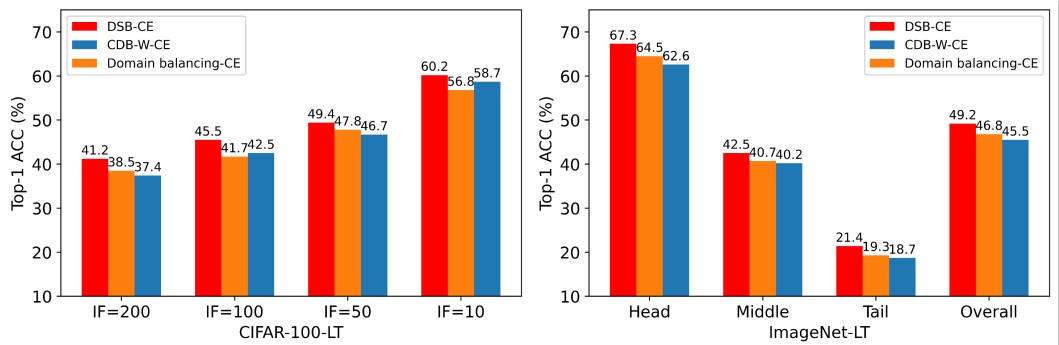

Figure 18: Accuracy comparison with other methods of measuring class-level difficulty.

We implemented CDB loss [37], LOCE [14], and domain balancing [2] on ImageNet-LT. Observing Figure 18, our proposed semantic scale balanced learning outperforms these three approaches. In addition to the comparison with the class-level difficulty weighting method, we add the results of the improvements to PaCO [10] in Table 2. Balanced softmax is included in PaCO, and although we have shown in Table 2 that our method significantly improves Balanced softmax, we still improved PaCO and conducted experiments to allay researchers' concerns. All experiments adopted the same training strategy and parameters as in Table 2.

H.3    The performance of dynamic semantic-scale-balanced learning in generalized long-tailed learning

Invariant feature learning (IFL [56]) considers both inter-class long tail and intra-class long tail and further defines the **generalized long-tailed classification**. The intra-class long tail has not been considered before, and invariant feature learning takes it into account in the long-tailed classification problem for the first time, which is remarkable progress in solving the long-tailed problem. IFL decomposes the probabilistic model of the classification problem as $P(y \mid x) = \frac{P(x|y)}{P(x)}P(y)$ and defaults the class with few samples to be the weak class. **It should be noted** that our study found that the geometric properties of the manifolds corresponding to different class distributions $P(x)$ will

affect the classification difficulty, which breaks the previous perception, so the inter-class long-tail problem still has huge research potential. The existence of data manifolds is already a consensus, and data classification can be regarded as the unwinding and separation of manifolds. Typically, a deep neural network consists of a feature extractor and a classifier. Feature learning can be considered as manifold unwinding, and a well-learned feature extractor is often able to unwind multiple manifolds for the classifier to decode. In this view, all factors about the manifold complexity may affect the model's classification performance. Therefore, we suggest that future work can explore the inter-class long-tailed problem from a geometric perspective. **Also, both the inter-class long tail and the intra-class long tail need to be considered, which will greatly alleviate the long-tailed problem**.

Invariant feature learning estimates relatively unbiased feature centers by constructing the resampling strategy and uses center loss for unbiased feature learning. We applied IFL to dynamic semantic-scale-balanced learning to consider both the inter-class long tail and the intra-class long tail, and validated it on ImageNet-LT and iNaturalist2018. Experiments show that DSB-CE combined with IFL achieves further performance improvement, and we have supplemented the results and analysis in Table 2.

Table 11: Evaluation on MSCOCO-GLT.

| Protocols | | CLT | GLT | ALT |
|---|---|---|---|---|
| < **Accuracy** \| **Precision** > | | Overall | Overall | Overall |
| Re-balance | cRT [27] | 73.64 \| 75.84 | 64.69 \| 68.33 | 49.97 \| 50.37 |
| | LWS [27] | 72.60 \| 75.66 | 63.60 \| 68.81 | 50.14 \| 50.61 |
| | Deconfound-TDE [55] | 73.79 \| 74.90 | 66.07 \| 68.20 | 50.76 \| 51.68 |
| | BLSoftmax [43] | 72.64 \| 75.25 | 64.07 \| 68.59 | 49.72 \| 50.65 |
| | BBN [80] | 73.69 \| 77.35 | 64.48 \| 70.20 | 51.83 \| 51.77 |
| | LDAM [4] | 75.57 \| 77.70 | 67.26 \| 70.70 | 55.52 \| 56.21 |
| | DSB-LDAM | 76.63 \| 78.95 | 68.15 \| 71.87 | 56.16 \| 56.87 |
| | BLSoftmax + IFL [56] | 73.72 \| 77.08 | 64.76 \| 70.00 | 52.97 \| 53.52 |
| | DSB-BLSoftmax | 73.96 \| 77.37 | 65.03 \| 70.15 | 50.24 \| 51.36 |
| | DSB-BLSoftmax + IFL | 74.64 \| 78.06 | 65.47 \| 70.83 | 53.08 \| 53.75 |
| | cRT + IFL [56] | 76.21 \| 79.11 | 66.90 \| 71.34 | 52.07 \| 52.85 |
| | DSB-cRT | **76.82** \| **79.95** | **67.26** \| **71.73** | 51.41 \| 51.94 |
| | LWS + IFL [56] | 75.98 \| 79.18 | 66.55 \| 71.49 | 52.07 \| 52.90 |
| | DSB-LWS | **76.55** \| **80.06** | **67.03** \| **72.15** | 51.64 \| 51.16 |

We note that IFL proposes two datasets ImageNet-GLT and MSCOCO-GLT and three testing protocols. Since our paper already contains a large number of experiments, we selected to conduct experiments on MSCOCO-GLT with the same experimental settings as IFL. The results are shown in Table 11. On the CLT and GLT protocols, we significantly improve the performance of BL-softmax, LDAM, and BL-softmax+IFL. Also, our approach promotes the performance of the above three methods on the ALT protocol, which may be caused by the additional gain from stronger inter-class discriminability. Due to page limitations, the experiment is tentatively supplemented in the appendix, and we will include this experiment in the main text if the paper is accepted. The experiments show that alleviating both inter-class long tail and intra-class long tail can significantly improve the model performance, so **we encourage researchers to pay attention to the intra-class long-tailed problem**.

## H.4 Performance of Dynamic Semantic-Scale-Balanced Learning on Three Subsets of ImageNet

We divided ImageNet into Head, Middle, and Tail subsets based on semantic scale, which contain 333, 333, and 334 classes, respectively. The performance of DSB-CE and CE on the three subsets when the backbone networks are VGG-16 and ResNet-18 is shown in Figure 19.

The experimental results show that semantic-scale-balanced learning significantly improves the performance of CE on Tail subset. Meanwhile, DSB-CE also outperforms CE on Head and Middle, which may be caused by the performance gain from better feature learning. In addition to the classification problem, we hope to introduce semantic scale imbalance in the fields of object detection, semantic segmentation, etc. to promote the fairness of the model.

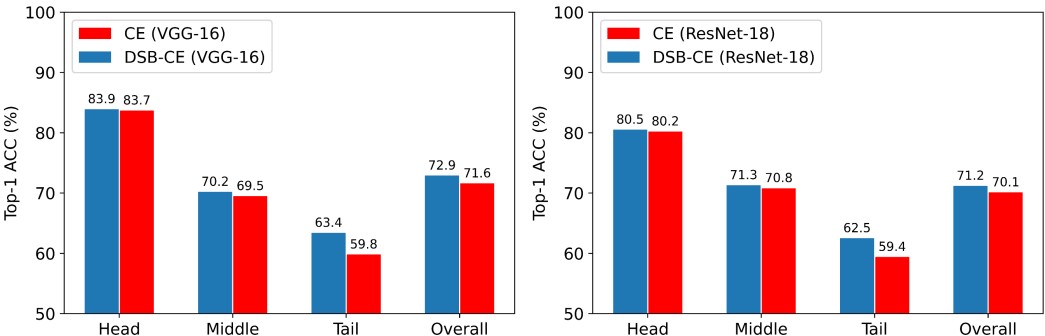

Figure 19: Comparison of CE and DSB-CE on ImageNet, where the backbone networks are VGG-16 and ResNet-34, respectively. It can be observed that dynamic semantic-scale-balanced learning significantly improves the tail class performance.

# I   APPLY SEMANTIC SCALE TO SOLVE OTHER PROBLEMS

## I.1   SELECT WELL-REPRESENTED DATA

Downsampling the head class is one of the methods to alleviate the long tail problem, which balances the number of samples but leads to the loss of head class information. Therefore, it is important to develop a downsampling method that preserves the head information. We propose an idea to select well-represented data based on the geometric meaning of semantic scale.

The existence of data manifolds is a consensus that the same class of data is usually distributed around a low-dimensional manifold. Different dimensions of the manifold represent different physical characteristics, and samples located at the edges of the manifold often tend to overlap with other manifolds. Therefore, we believe that the following two principles should be obeyed when downsampling:

- Uniform sampling inside the manifold. It ensures that the volume of the manifold does not shrink significantly after downsampling.

- Increase the sampling rate of samples at the edges of the manifold. It makes the sampled distribution with significant bounds, which helps to improve the robustness of the classification model.

As shown in Figure 20, we refer to the strategy that obeys the above sampling principles as **"pizza" sampling**.

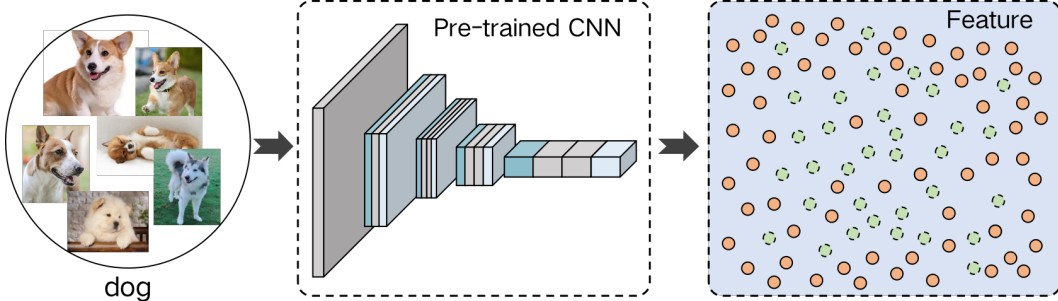

Figure 20: Schematic diagram of pizza sampling. Yellow samples indicate the selected samples and green samples indicate the discarded samples.

Uniform sampling is easy to do, but how do we sample as many samples as possible from the edges of the manifold? We propose to randomly sample $k$ subsets in the original sample set and calculate the semantic scales of the subsets. Then repeat the above operation several times and select the subsets with the largest semantic scales as the final samples.

### I.2 Guide Data Collection

When collecting data that has never been studied before, we do not know how many samples to collect to represent their corresponding class well because of the lack of prior knowledge. When too few samples are collected, the class is under-represented. And sampling too many samples will consume huge costs. The marginal effect of the semantic scale can help us judge whether the currently collected samples have enough feature diversity, and we can stop collecting samples when the feature diversity tends to be saturated. Specifically, the data collection process is as follows.

(1) For class $c$, $m$ samples are collected each time.

(2) After the $(n-1)$th collection of samples, there are $(n-1) \times m$ samples, and the semantic scales of these samples are calculated.

(3) After the $n$th collection of samples, there are $n \times m$ samples, and the semantic scales of these samples are calculated.

(4) Calculate the increment of the semantic scale for the $n$th time relative to the $(n-1)$th time.

(5) Calculate $\frac{(S_n - S_{n-1})}{S_n}$. If the increment of semantic scale is less than $\alpha\%$ of $S_n$, it means that the feature diversity of class $c$ has not changed significantly and the sample collection can be stopped. The parameter $\alpha$ can be adjusted according to the needs of the task.

Geometric analysis of data manifolds can bring new perspectives to data science. We will open source the toolkit for measuring the information geometry of data, which includes the application of semantic scale in various scenarios, such as data collection, and representative data selection.

## J More explanation of Figure 2

To see it more clearly, we zoomed in on Figure 2 and plotted it in Figure 24. Previous studies have observed that (1) given sufficient data, the classification performance gain is marginal with additional samples. (2) When the data is insufficient, the classification performance drops sharply as the number of training samples decreases. We speculate that phenomenon 1 may be caused by the marginal effect of feature diversity. It should be noted that CB loss considers marginal effects, but it only qualitatively describes the gradual flattening of feature diversity with the increasing number of samples. Taking CIFAR-10 as an example, we first select a few samples for each class, train the model and test the accuracy. Then new samples are continuously added to the original samples instead of re-selecting more samples to train the model. The experiments corresponding to each point in Figure 2 are trained from scratch. While increasing the data we find that there are marginal effects of semantic scale, which indicates that our proposed measurement is as expected. The marginal effects of feature diversity explain phenomenon 1.

However, phenomenon 2 is not explained by the marginal effects, and the effective number of samples from CB loss does not predict phenomenon 2 at all, because the effective number of samples does not grow faster than the number of samples (which we have analyzed in Section 2). We experimentally find that when the samples are few, the feature diversity measured by the semantic scale increases rapidly with the number of samples, and this increase is faster than the linear increase. The rapid increase of feature diversity measured by the semantic scale explains phenomenon 2.

## K Can the semantic scale capture the hierarchical structure?

HCSC [15] constructs the hierarchical structure of classes by bottom-up k-means, and we use the example shown by HCSC to validate our approach. Given the following seven classes: Poodles, Samoyeds, Labradors, Persian, Siamese, Chimpanzee, and Gorilla, each class contains $1,000$ samples, and the hierarchical structure of the seven classes is shown in Figure 25.

We collect $1,000$ images for each of the three parent classes (Dogs, Cats, and Monkeys), which can adequately represent the three parent classes, i.e., the feature richness is sufficient. **Then can the semantic scale be used to match the correct parent classes for the seven classes?** According to our theory, the manifolds of the child classes should be in the manifold of the corresponding parent class, and they have an inclusion relationship. Therefore, when the data of the child classes are mixed

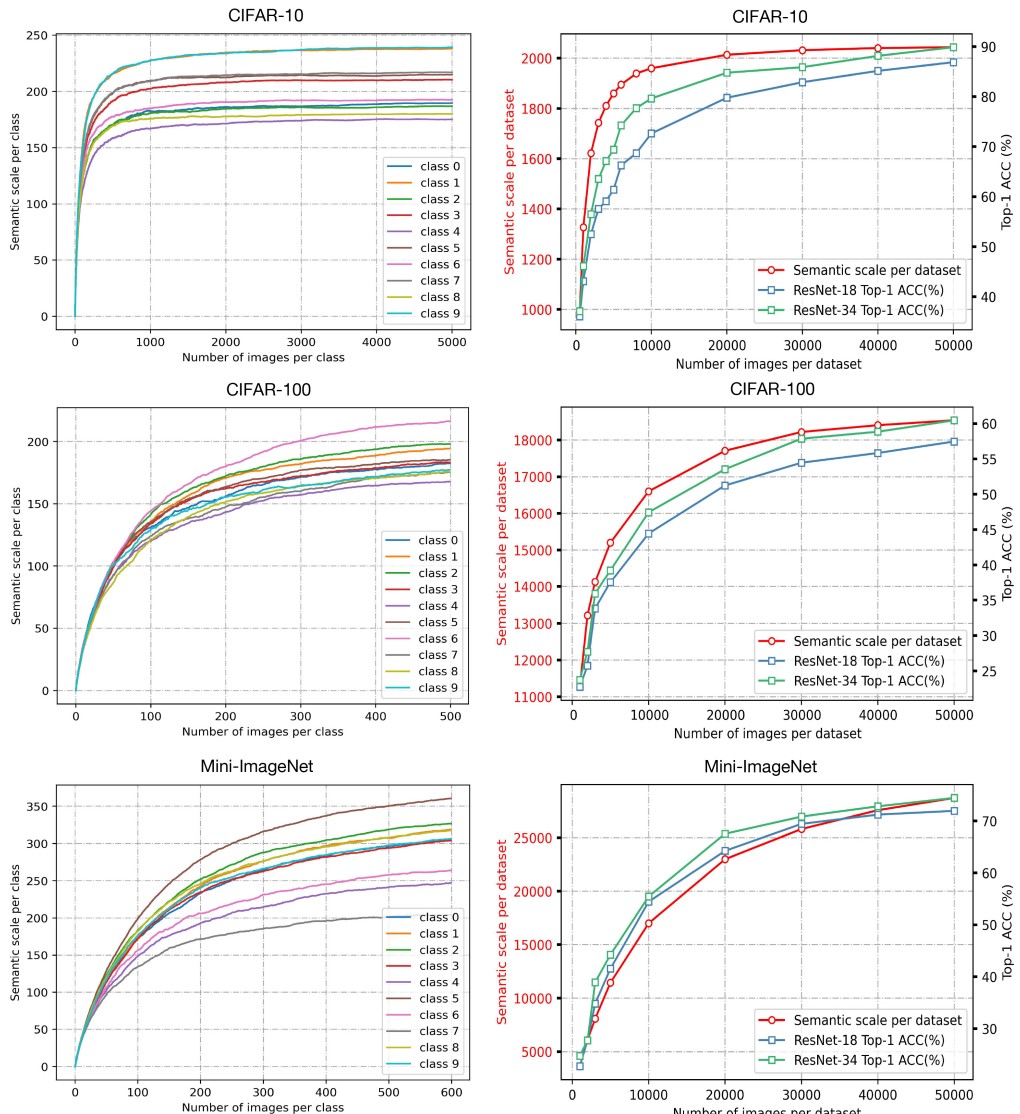

Figure 21: **Left column**: curves of semantic scales with increasing number of samples for the first ten classes from different datasets. **Right column**: for different sub-datasets, curves of the sum of semantic scales for all classes and top-1 accuracy curves of trained ResNet-18 and ResNet-34. All models are trained using the Adam optimizer [28] with an initial learning rate of $0.01$ and then decayed by $0.98$ at each epoch.

into the data of the parent class, the manifold volume of the parent class will not change significantly. We propose the matching method of semantic hierarchy based on this property. The specific steps are as follows.

(1) train a ResNet-18 classification model on seven child classes. We set the batch size to $64$ and adopt the adam optimizer with a learning rate of $0.01$ (linear decay), a momentum of $0.9$, and a weight decay factor of $0.005$.

(2) Extract the features of all samples from seven child classes and three parent classes.

(3) Calculate the semantic scales of the three parent classes.

(4) Select a child class $c$ from the seven child classes.

(5) Mix the data of child class $c$ into the data of each parent class and calculate the semantic scale of the mixed data, we can get three values.

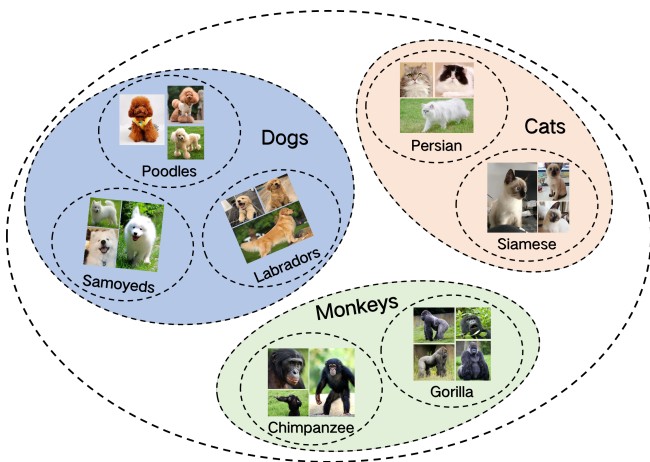

Figure 22: Image datasets typically contain multiple semantic hierarchies.

Table 12: **Results of matching parent class for each child class**, where the Ratio of semantic scales denotes the ratio of the semantic scales of the parent class after mixing to before mixing, Predicted parent class means the parent class we matched for the child class, and Real parent class denotes the real parent class corresponding to the child class.

| child class | Poodles | | | Samoyeds | | | Labradors | | | Persian | | |
|---|---|---|---|---|---|---|---|---|---|---|---|---|
| parent class | Dogs | Cats | Monkeys | Dogs | Cats | Monkeys | Dogs | Cats | Monkeys | Dogs | Cats | Monkeys |
| Ratio of semantic scales | 1.06 | 1.72 | 1.94 | 1.03 | 1.68 | 1.89 | 1.05 | 1.64 | 1.83 | 1.75 | 1.08 | 1.87 |
| Predicted parent class | ✓ | | | ✓ | | | ✓ | | | | ✓ | |
| Real parent class | ✓ | | | ✓ | | | ✓ | | | | ✓ | |

| child class | Siamese | | | Chimpanzee | | | Gorilla | | |
|---|---|---|---|---|---|---|---|---|---|
| parent class | Dogs | Cats | Monkeys | Dogs | Cats | Monkeys | Dogs | Cats | Monkeys |
| Ratio of semantic scales | 1.69 | 1.02 | 1.84 | 1.87 | 1.83 | 1.06 | 1.82 | 1.89 | 1.02 |
| Predicted parent class | | ✓ | | | | ✓ | | | ✓ |
| Real parent class | | ✓ | | | | ✓ | | | ✓ |

(6) Calculate the changes in the semantic scales of the three parent classes and sort them.

(7) Match the parent class with the smallest change in semantic scale for child class $c$.

(8) Perform steps (3) to (7) for the remaining six child classes.

We summarize the ratio of the semantic scales of the parent classes after mixing to before mixing in Table 12. If the change in the semantic scale of a parent class is small after a child class is mixed into that parent class, they are considered to have a nested relationship. Based on the above method, we successfully match each child class to the parent class. Experimental results show that our proposed measure of semantic scales can capture the semantic hierarchy of classes. Our study can inspire hierarchical feature learning as well as facilitate its performance in downstream tasks.

## L    FUTURE WORK AND CHALLENGES

### L.1    MODEL-INDEPENDENT MEASURE OF DATA DIFFICULTY

The performance of the models varies across classes. In the past, it was believed that model bias was caused by an imbalance in sample numbers, but a growing body of research suggests that sample numbers are not the only factor affecting model bias. Of course, model bias is also introduced not by the model structure, but by the characteristics of the data itself that affect model performance. Therefore, it is very important to propose model-independent measurements to represent the data itself, and this work will greatly contribute to our understanding of deep neural networks. In this

paper, the effect of the volume of the data manifold on the model bias is explored from a geometric perspective. It provides a new direction for future work, namely the geometric analysis of deep neural networks. The geometric characteristics of the data manifold will help us further reveal how neural networks learn and inspire the design of neural network structures.

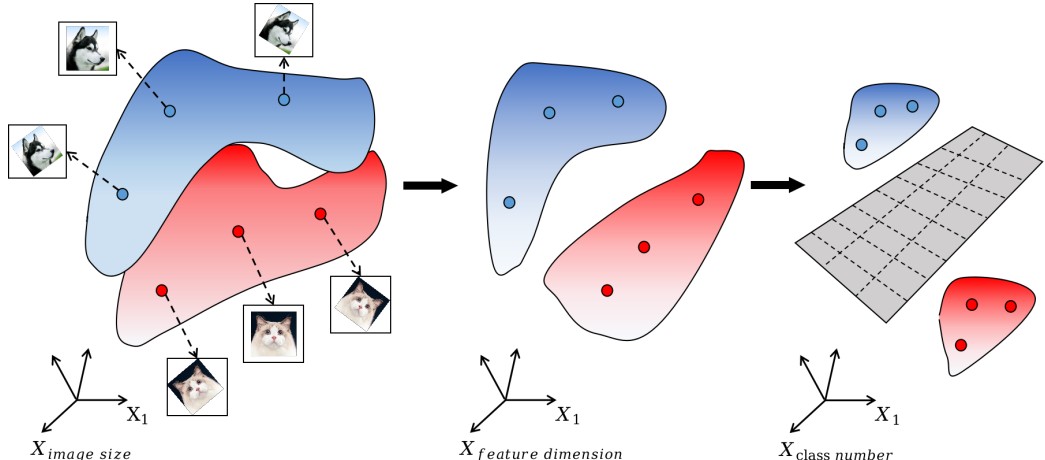

Figure 23: Changes in the geometry of data manifolds as they are transformed in a deep neural network. The classification process of the data includes untangling the manifolds from each other and separating the different manifolds.

## L.2 A GEOMETRIC PERSPECTIVE ON DATA CLASSIFICATION

Natural datasets have intrinsic patterns that can be generalized to the manifold distribution principle: the distribution of a class of data is close to a low-dimensional manifold. As shown in Figure 23, data classification can be regarded as the unwinding and separation of manifolds. When a data manifold is entangled with other perceptual manifolds, the difficulty of classifying that manifold increases. Typically, a deep neural network consists of a feature extractor and a classifier. Feature learning can be considered as manifold unwinding, and a well-learned feature extractor is often able to unwind multiple manifolds for the classifier to decode. In this view, all factors about the manifold complexity may affect the model's classification performance. Therefore, we suggest that future work can explore the inter-class long-tailed problem from a geometric perspective.

## L.3 INTRODUCE SEMANTIC SCALE IMBALANCE IN OBJECT DETECTION

Long-tailed distribution is one of the main difficulties faced by object detection algorithms in real-world scenarios. The classical object detection algorithms are generally trained on some manually designed datasets with relatively balanced data distribution. In contrast, the accuracy of these algorithms tends to suffer significantly on long-tailed distributed datasets. So far, methods for foreground-background imbalance and class imbalance have been proposed extensively, but these methods are based on the number of objects to define the degree of imbalance and cannot explain more phenomena. We will give examples below.

In the field of object detection, it is often encountered that although a class does not appear frequently, the model can always detect such instances efficiently. It is easy to observe that classes with simple patterns are usually easier to learn, even if the frequency of such classes is low. Therefore, classes with low frequency in object detection are not necessarily always harder to learn. We believe that it is a valuable research direction to analyze the richness of the instances contained in each class, and then pay more attention to the hard classes. The dimensionality of all images or feature embeddings in the image classification task is the same, which facilitates the application of the semantic scale proposed in this paper. However, the non-fixed dimensionality of each instance in the field of object detection brings new challenges, so we have to consider the effect of dimensionality on the semantic scale, which is a direction worthy of further study.

### L.4    CHALLENGES OF CLASS IMBALANCE IN DEEP LEARNING

Class imbalance remains a major challenge in the field of deep learning. Data imbalance classification, although widely studied, still lacks effective and clear methods and guidelines. The problem of object detection for class imbalance is still in its infancy and requires a greater investment of attention. In the following, we summarize the important future challenges and research directions in this field.

(1) *The more precise measure of class difficulty.* An increasing number of studies have shown that the sample number does not accurately reflect the accuracy of the model in recognizing classes. Therefore, more extensive measures should be proposed to redefine the long-tail distribution to facilitate classification and object detection tasks and further expand the scope of research on long-tailed recognition. For example, a dataset with perfectly balanced sample numbers may not be balanced under other measures.

(2) *Long-tailed distribution of properties in classes.* As shown in Figure 24 [56], previous studies have focused on the imbalance between classes and ignored the imbalance of properties within each class. For example, most pandas have black and white fur, and only a small proportion of pandas are brown. In visual recognition tasks, we should not only pursue the overall accuracy of the class but also pay attention to whether samples with sparse properties in a class can be classified accurately. In medical image classification, the above point is particularly important. For example, pulmonary diseases contain many different types of diseases, and generally the more severe the disease tends to have a smaller sample number, suggesting that there is an imbalance of properties under the label of pulmonary disease. We hope to be able to recognize more severe diseases more accurately so that patients do not miss the best time to treat them.

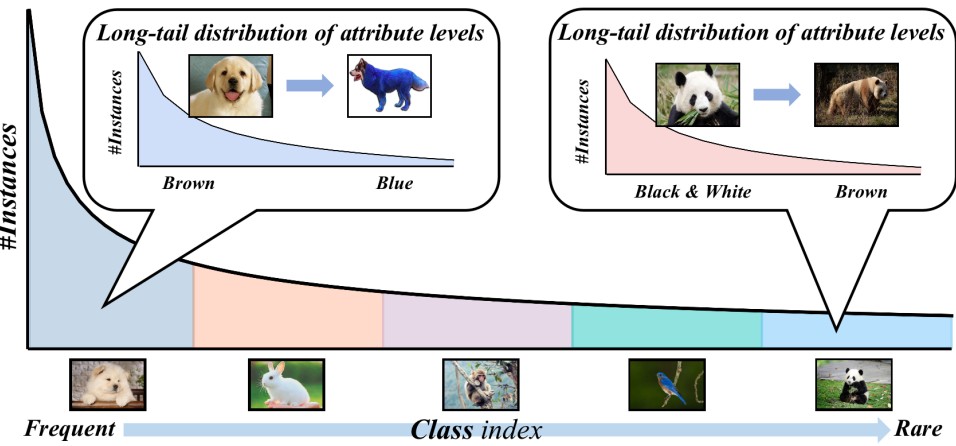

Figure 24: Class-level long-tailed distribution and intra-class attribute long-tailed distribution.

(3) *Generalization performance of the model outside the training domain of the tail class.* As shown in Figure 25, tail classes often have very few samples, so these samples do not well represent the true distribution of the tail classes, which results in the model consistently failing to learn and adapt to the tail classes correctly. Obviously, recovering the underlying distribution of the tail classes helps the generalization performance of the model outside the training domain of the tail classes. It is currently shown that similar classes have similar distribution statistics (variance), which can lead researchers to recover the underlying distribution of tail classes. However, the current research is still in its infancy, and it is not a sufficiently stringent assumption that similar classes have similar variances. Therefore, we hope that in the future researchers will be able to help recover the true distribution of tail classes by more means.

(4) *How to choose the appropriate long-tailed recognition method in the task.* Up to now, a large number of visual recognition methods on long-tail distribution have been proposed. While individual methods have positive performance in long-tailed recognition tasks, some combinations of methods may have negative effects. Few studies have focused on the

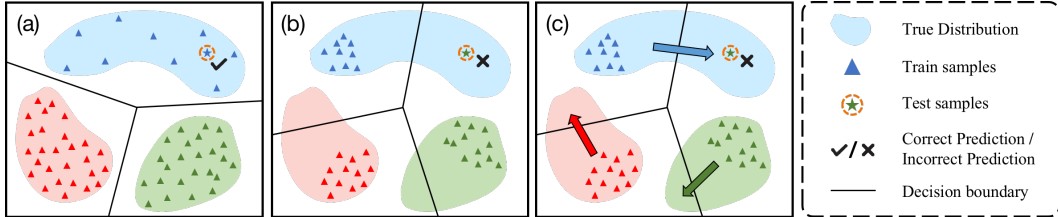

Figure 25: (a) When the samples uniformly cover the true data distribution, the model can learn the correct decision boundaries and can correctly classify unfamiliar samples to be tested. (b) When the samples cover only a portion of the true distribution, unfamiliar samples to be tested are highly likely to be misclassified due to the error in the decision boundary. (c) The direction in which the arrow points is the best direction to expand the sample.

selection and combination of different training techniques and methods. In the future, it is possible to explore how to select existing methods on specific tasks, and further, effective combinations of different methods are important.

(5) *Multi-domain deep long-tailed learning*. Past research has typically focused on the problem of long-tailed distribution over a single domain, which has limited the research ideas. As shown in Figure 26, data from multiple domains can complement each other to alleviate the long-tailed distribution of classes [73]. For example, in plant and animal classification, cameras are placed in different places to capture animals, but some animals only appear in a fixed area, which leads to different label distributions for animals captured by different cameras. But by combining the data from all cameras, a more balanced class can be obtained. Similarly, a similar situation occurs in other practical applications. For example, in a visual recognition problem, the few classes from "photo" images can be complemented by a potentially rich sample from "sketch" images. In autonomous driving, a few classes of "real" life accidents can be enriched by accidents generated in "simulations". In addition, in medical diagnosis, data from different populations can be mutually augmented, e.g., a small sample from one institution can be combined with the majority of possible instances from other institutions. In these examples, different data types can act as different domains, and such multi-domain data can also be utilized effectively to address data imbalances.

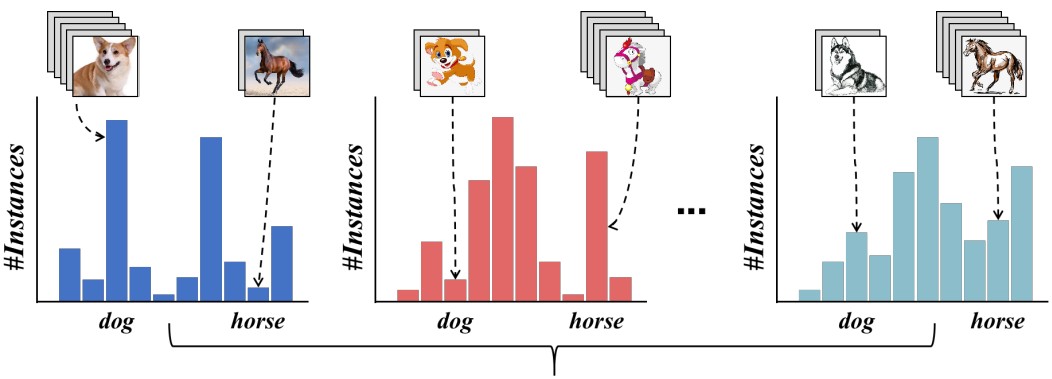

Figure 26: The frequency of the same class appearing in different domains may differ significantly, assuming a smaller sample of horses in the real world and a larger number of horses in cartoon form. Images from different domains can complement each other to form a dataset with a balanced sample number. The purpose of multi-domain deep long-tailed learning is to train unbiased models using data from multiple domains and generalize over all domains.

(6) *Recognition of unbalanced data streams*. Continuous learning aims to process new data that is continuously generated in order to dynamically update and adapt the model to the latest data domain. Challengingly, as new data is generated, the degree of imbalance between classes changes and what used to be a tail class may become a head class. The long-tailed distribution of properties within classes can also affect the performance of the model if

concept drift occurs. Thus the key to handling unbalanced data streams is to evaluate the class-level difficulty and the long-tailed distribution of properties within classes in real time, which is a huge challenge.

(7) *Augmentation methods for other modally unbalanced data.* Methods for multi-sample synthesis are widely used in image data augmentation, such as Mixup and Cutout, but there is still a lack of data augmentation methods for other modal data (e.g., speech and table). Researchers can design a more general method to generate samples of any type of data.

(8) *Other long-tailed visual recognition tasks.* Current research focuses on long-tail image classification, while less attention has been paid to long-tail object detection, image segmentation, and regression tasks. In object detection, there are multiple imbalances, such as foreground-background imbalance and imbalance between classes belonging to the foreground, which are unresolved challenges. With further applications of deep learning, research on imbalance learning in various fields will be of great benefit for real-world applications.

This study suggests some future avenues of inquiry to further deepen and expand the study of unbalanced learning. Of course, the scope of future inquiry into unbalanced learning is not limited to the eight challenges mentioned above, and we believe that new questions will arise in the course of inquiry into these eight challenges, but that researchers will eventually address them over time.

*Science is endless, it is an eternal mystery.*

Albert Einstein

