# OpenReview forum: "Delving into Semantic Scale Imbalance"
_ICLR.cc/2023/Conference — ICLR 2023 poster_

### Official Review · Reviewer_bNtG · 2022-10-17

**Confidence:** 4
**Correctness:** 3
**Technical Novelty And Significance:** 3
**Empirical Novelty And Significance:** 3
**Recommendation:** 6

**Clarity, Quality, Novelty And Reproducibility:**

- The paper is easy to follow.
- The quality is overall good but there are some issues should be resolved.
- The novelty is fair.
- The reproducibility: the source code is expected to be released.

**Strength And Weaknesses:**

Strengths:
1. The problem of class imbalance is important for practical applications.
2. It is interesting and reasonable to rethink class imbalance beyond just the sample number per class.
3. The formulated semantic scale is new to me.

Weaknesses:
1. The designed DSB loss relies not only on the semantic scale but also the inter-class interference. Therefore, it is unclear how much performance gains are from the semantic scale and how much is from the inter-class interference. Since the main innovation is the semantic scale, it is better to show its empirical effectiveness.
2. In Sec 4.1, this paper argues that "all methods that rebalance the loss function or adjust the sampling rate based on the sample number can be improved with the semantic scale, since both are natural measures and they are not model-dependent." This argument is not clear enough. First of all, the evaluation of the semantic scale and the inter-class interference are based on model features, so they are model-dependent. Moreover, in my view, combining the proposed reweighting loss and existing class re-balancing is not that easy, since they are competing with each other in reversing imbalance to some degree. Therefore, the combination pipeline should be carefully designed and the hyper-parameters should be carefully tuned; otherwise, the model performance may not be improved and even become worse.
3. There are many re-weighting long-tailed approaches that do not rely on the number of samples, like Uncertainty-based margin learning [A], LOCE [B] and Domain balancing [C]. Please discuss some of them if you have not done so and also compare the proposed method with them to show superiority.
[A] Striking the right balance with uncertainty. In CVPR 2019.
[B] Exploring classification equilibrium in long-tailed object detection. In ICCV 2021.
[C] Domain balancing: Face recognition on long-tailed domains. In CVPR 2020.
4. Can the proposed semantic scale be used to guide re-sampling for long-tailed data? Whether is it better guidance than the sample number per class for class-balanced re-sampling?
5. The performance of BS [39] on ImageNet-LT is not that bad. Many recent studies have verified this. Please recheck your implementation in the experiment.
6. In Section 5.2, can you order the semantic scale of different classes, and group them into three groups? I wonder how many accuracies of different groups are improved. Whether are the classes with small semantic scales improved more?

**Summary Of The Paper:**

This paper studies class imbalance and proposes that we should model the imbalance based on the semantic scale per class instead of the number of samples per class. Following this, this paper conducts a series of empirical studies to verify this idea and proposes a new method to handle class imbalance. Empirical results demonstrate the effectiveness of the proposed method.

**Summary Of The Review:**

This paper is overall good, but there are several problems that need to be resolved.

*******************
Post-rebuttal: Thanks for the response and modifications. The quality of this paper is further improved now. However,  I still think the proposed method is a mix of different ideas, considering that inter-class interference is not that important. Empirical analysis and suggestions regarding inter-class interference are good, but it is not that elegant when designing them into one method. Even so, I appreciate the efforts of the authors and increase my rate to 6.

---

> ### Author Response · Authors · 2022-11-13
> **Response to Reviewer bNtG.**
>
> Dear Reviewer bNtG,
>
> &nbsp;
>
> Thank you very much for your insightful and visionary comments, which helped us to improve the paper.
>
> **Weakness 1:** We have weakened the effect of inter-class interference when designing the measurement of semantic scale imbalance. The Pearson correlation coefficients between class accuracy and inter-class interference, semantic scale, and semantic scale considering inter-class interference, respectively, are shown in Table 1. It can be seen that the Pearson correlation coefficient between semantic scale and class accuracy on CIFAR-10-LT without considering inter-class interference still reaches 0.8688, while the correlation coefficient between inter-class distance W and class accuracy is only 0.2957, which illustrates the importance of semantic scale. In addition, we have added the correlation coefficients between the effective sample numbers and class accuracies in Table 1. It can be observed that the correlation between effective sample number and class accuracy is almost the same as the correlation between sample number and class accuracy, which is due to the fact that effective sample number is a monotonic function of sample number. To demonstrate the performance of dynamic semantic-scale-balanced learning without considering inter-class interference, we conducted experiments on ImageNet-LT and supplemented them in **Appendix H.1**. It can be observed that dynamic semantic-scale-balanced learning without considering inter-class interference still has a strong performance.
>
> **Weakness 2:** We would like to express that there are various types of methods in deep long-tailed learning, such as ensemble learning (RIDE, TADE, etc.), decoupled training, and cost-sensitive learning, etc. Our approach can replace the weights in cost-sensitive learning and also improve other methods. Several types of methods are improved in the experiments, such as LDAM, BLS, LADE, and RIDE, which demonstrates the generality of our approach.
>
> **Weakness 3:** We note a recent work (CDB loss) published in IJCV that measures class difficulty. In addition, domain balancing also measures class-level difficulty, so we compare semantic-scale-balanced learning with them. The introduction and comparison experiments of the above two methods are shown in **Appendix H.2**. The experiments indicate that our approach significantly outperforms the other methods.
>
> **Weakness 4:** Your question is very visionary and valuable. It is very important to avoid losing information about the head classes when downsampling them, so we propose an idea to select well-represented data based on the semantic scale, which aims to represent the original distribution as much as possible with fewer samples. Our proposed semantic scale has great potential for application. In engineering applications, how many samples should be collected for each class is the most appropriate? When too few samples are collected, the class is under-represented, while too many will consume huge costs. Our approach can effectively solve this problem by stopping the collection when the semantic scales tend to be saturated. The details are supplemented in **Appendix I**.
>
> **Weakness 5:** The main contribution of Balanced meta-softmax is to propose a meta-learning-based sampling method to estimate the optimal sampling rate for different classes in unbalanced learning, and it also includes Balanced softmax. To demonstrate the effect of re-weighting with semantic scale, we only dynamically re-weight the Balanced softmax for evaluation. Therefore, the experimental results are accurate.
>
> **Weakness 6:** Thank you very much for your suggestion which led us to a more detailed analysis. In **Appendix H.4**, we divide ImageNet into Head, Middle and Tail subsets based on semantic scale, which contain 333, 333 and 334 classes, respectively. The performance of DSB-CE and CE on the three subsets when the backbone networks are VGG-16 and ResNet-18 is shown in Figure 15. The experimental results show that semantic-scale-balanced learning significantly improves the performance of CE on Tail subset. Meanwhile, DSB-CE also outperforms CE on Head and Middle, which may be caused by the performance gain from better feature learning.
>
> Thank you very much for your review, we look forward to hearing from you and wish you happy work.
>
> &nbsp;
>
> Yours sincerely,
>
> Authors of Paper 42

---

> > ### Comment · Reviewer_bNtG · 2022-11-18
> > **Further discussion**
> >
> > Thanks for the response. It addresses most of my concerns, and some new analyses further improve the quality of the paper. However, some questions still exist. First, According to Figure 13, inter-class interference only has a very limited contribution. Then, why do you use it in the proposed method? **Considering that inter-class interference is not important for the proposed method, it is better to remove the unnecessary component and keep the proposed technique as simple and focused as possible**. Regarding question 2, I see your point but your argument is too strong. The results show that your proposed method is beneficial for some of the long-tailed methods but not all. It is better to soften the argument of  "all methods that rebalance the loss function or adjust the sampling rate based on the sample number can be improved with the semantic scale". Regarding question 5, Table 2 shows that using BS to train **ResNeXt50** on ImageNet-LT obtains 40% overall accuracies, and your method can improve it by 2% accuracy gains. However, a recent survey of long-tailed learning (https://arxiv.org/pdf/2110.04596.pdf) shows that the same baseline can achieve more than 50% accuracy. Although it is not a major issue, it is better to implement these baselines to their best.

---

> > > ### Author Response · Authors · 2022-11-19
> > > **Further Response to Reviewer bNtG**
> > >
> > > Dear Reviewer bNtG,
> > >
> > > &nbsp;
> > >
> > > Thank you very much for reviewing our response in detail, and your latest suggestions make our paper more rigorous.
> > >
> > > Although the experiments on ImageNet-LT show that inter-class interference is not particularly important for the proposed approach, it is also important to point out that inter-class interference is also a factor that affects model bias, and we would like to alert more researchers to this point, and therefore prefer to keep this part.
> > >
> > >
> > > For question 2, following your comments, the original text was modified to “Our approach has great potential to improve the methods of re-balancing loss and adjusting sampling rate based on the number of samples, because both the semantic scale and the number of samples are natural measures and they are not model-dependent.”
> > >
> > > We re-implemented the state-of-the-art BS, and the latest experimental results were revised in Table 2. If you have questions, you can feel free to contact us.
> > >
> > > &nbsp;
> > >
> > > Yours sincerely,
> > >
> > > Authors of Paper 42

---

> > > > ### Comment · Reviewer_bNtG · 2022-11-21
> > > > **Response to authors**
> > > >
> > > > Thanks for the response. The paper is better now. I still think the proposed method is a mix of different ideas, considering that inter-class interference is not that important. Empirical analysis and suggestions are good, but it is not that elegant when designing them into one method. Even so, I appreciate the efforts of the authors and will increase my rate to 6.

---

> > > > > ### Author Response · Authors · 2022-11-21
> > > > > **Thank You！**
> > > > >
> > > > > Dear Reviewer bNtG,
> > > > >
> > > > > &nbsp;
> > > > >
> > > > > Thank you very much for your constructive comments, which make our paper better again and again. We agree with your suggestion and will emphasize in the final version of the paper that the approach is more concise and efficient when inter-class interference is ignored.
> > > > >
> > > > > We wish you good health and happy work.
> > > > >
> > > > > &nbsp;
> > > > >
> > > > > Yours sincerely,
> > > > >
> > > > > Authors of Paper 42

---

### Official Review · Reviewer_DeDZ · 2022-10-19

**Confidence:** 4
**Correctness:** 3
**Technical Novelty And Significance:** 3
**Empirical Novelty And Significance:** 3
**Recommendation:** 6

**Clarity, Quality, Novelty And Reproducibility:**

The topic is relative novel, since the work from the same topic is just conducted in the same year and in a few months ago.

[Posthoc Response]
The code of this work can be open to promote the further explorations in this direction.

**Strength And Weaknesses:**

In summary, there are some positive points regarding this work, summarized as follows,

(1) The topic is relatively novel and meaningful in the perspective of long-tailed learning, since the quantity bias is not the essential factor to decide the performance of imbalance learning.

(2) The proposed measure for imbalance degree is effective, which is demonstrated by a range of validation on uniform datasets and long-tailed datasets. Specially, the trend of the curve is approximately similar to the final performance, which is heuristic to future extensions.

(3) The proposed semantic-scale-based balanced loss has been demonstrated useful to a range of previous methods to improve the performance in both class-imbalanced and class-balanced cases.

However, there are also some clear flaws in terms of the current submission, which needs to be further improved.

(1) It is confusing that the authors discuss both the marginal effect and semantic-scale imbalance. Especially, the marginal effect is not the unique characteristic of imbalance learning, and also exists in the absolutely balanced learning. When the samples are sufficient to make the model reach the optimal, more samples might not bring gains.

(2) In terms of the semantic-scale-based imbalance learning, it is the approximately same topic with the recent work termed as the generalized long-tailed learning [1], which takes into the fine-grained pattern imbalancedness into account. The authors missed this work and did not discuss their difference as well as the corresponding comparison in the experimental part. Especially, they have released a benchmark in the github repository, which can be used in this work.

(3) The evaluation in the quantity-uniform datasets like CIFAR10 and CIFAR100 is ill-posed, since both the training and the test are IID. The improvement is not convincing to attribute to the imbalance learning, and conversely it is more possible to be explained as the effect of the hard negative mining by the re-weighting. Here, the authors need to carefully discuss the improvement in the IID instead of the Non-IID case about the training and the test.

(4) There are some concerns about the experiments, since it seems that the authors do not demonstrate the model can achieve the better improvement on the basis of the state-of-the-art method like PaCO and LA, which have adjusted the model according to the quantity imbalance. Note that, RIDE is a multi-branch ensemble model, and it is meaningless to show the improvement on RIDE achieves the state-of-the-art, since PaCO can also integrate this mechanism to achieve better performance. It will be better to conduct some experiments to compared LA or PaCO and your method in the same backbone, which can show the gap of the improvement between the explicit quantity bias and the implicit semantic-scale bias.  Again, it is better to have some comparison in the benchmark of [1].

[1] Invariant Feature Learning for Generalized Long-Tailed Classification. ECCV 2022.

**Summary Of The Paper:**

This work studies long-tailed learning in the semantic level, which goes deeper beyond the conventional quantity bias. Specially, the authors explore to explain multiple phenomena by defining the semantic level imbalance and propose a measure by volume of manifold to reweight the learning objective, which has been demonstrate to improve the performance effectively.

**Summary Of The Review:**

In summary, the current work has both pros and cons. I advise the authors to carefully consider the advices in the review and I can consider to raise the score if the concerns have been well solved.

---

> ### Author Response · Authors · 2022-11-13
> **Response to Reviewer DeDZ.**
>
> Dear Reviewer DeDZ,
>
> &nbsp;
>
> Thank you very much for your constructive comments, which enabled us to polish the paper.
>
> **Weakness 1:**
>
> It is a consensus that feature diversity has a marginal effect, and thus the marginal effect of the semantic scale suggests that our measurement of feature diversity is appropriate. In addition, the marginal effect of semantic scale also explains two phenomena pointed out in the abstract: **(1)** Given enough data, the classification performance gain is marginal with additional samples. **(2)** Classification performance decays precipitously as the number of training samples decreases when there is insufficient data.
>
> **Weakness 2:**
>
> Invariant feature learning (IFL) considers both inter-class long tail and intra-class long tail and further defines the generalized long-tailed classification. The intra-class long tail has not been considered before, and invariant feature learning takes it into account in the long-tailed classification problem for the first time, which is remarkable progress in solving the long-tailed problem. However, it should be noted that IFL's view on inter-class long tail is not different from previous studies, still assuming that classes with fewer samples are weak classes, while our approach breaks the previous perception. Our proposed semantic scale imbalance is an inter-class imbalance measure that can also consider the intra-class long-tailed problem. The discussion of invariant feature learning has been supplemented in **Appendix H.3**.
>
> Invariant feature learning estimates relatively unbiased feature centers by constructing the resampling strategy and uses center loss for unbiased feature learning. We applied IFL to dynamic semantic-scale-balanced learning to consider both the inter-class long tail and the intra-class long tail, and validated it on ImageNet-LT and iNaturalist2018. Experiments show that DSB-CE combined with IFL achieves further performance improvement, and we have supplemented the results and analysis in Table 2.
>
> **Weaknesses 3 and 4:**
>
> In **Appendix H.4**, We divided ImageNet into Head, Middle, and Tail subsets based on semantic scale, which contain 333, 333, and 334 classes, respectively. The performance of DSB-CE and CE on the three subsets when the backbone networks are VGG-16 and ResNet-18 is shown in Figure 15. The experimental results show that semantic-scale-balanced learning significantly improves the performance of CE on Tail subset. You suggest that this may be the effect of the hard negative mining, which also precisely shows that our approach can effectively mine the hard classes. In addition, we compare the semantic scale with other class difficulty measures in **Appendix H.2**, and the experiments indicate that our approach significantly outperforms the other methods.
>
> We improve a variety of classical methods in the experimental part, and due to the heavy workload, it is not possible to improve and experiment with all the methods. In fact, the Balanced softmax is included in PaCO, and we have significantly improved its performance in Table 2. To reassure you, we improve PaCO and conduct experiments, and the results are supplemented in Table 2. Since RIDE with multiple experts can be improved and PaCO can also be improved, we can certainly improve PaCO with multiple experts.
>
> We note that IFL proposes two datasets ImageNet-GLT and MSCOCO-GLT and three testing protocols. Since our paper already contains a large number of experiments, we selected to conduct experiments on MSCOCO-GLT with the same experimental settings as IFL. The results are shown in **Table 11**. On the CLT and GLT protocols, we significantly improve the performance of BL-softmax, LDAM, and BL-softmax+IFL. Also, our approach promotes the performance of the above three methods on the ALT protocol, which may be caused by the additional gain from stronger inter-class discriminability. Due to page limitations, the experiment is tentatively supplemented in **Appendix H.3**, and we will include this experiment in the main text if the paper is accepted. The experiments show that alleviating both inter-class long tail and intra-class long tail can significantly improve the model performance, so we encourage researchers to pay attention to the intra-class long-tailed problem.
>
> **Additional response:**
>
>  We show more experimental results and analysis in **Appendix H**. Potential applications of our approach to other problems are presented in **Appendix I**. Future research directions and how to introduce semantic scale imbalance into object detection are discussed in **Appendix J**.
>
> Thank you very much for your review, we look forward to hearing from you and wish you happy work.
>
> &nbsp;
>
> Yours sincerely,
>
> Authors of Paper 42

---

### Official Review · Reviewer_kLsB · 2022-10-20

**Confidence:** 4
**Correctness:** 3
**Technical Novelty And Significance:** 3
**Empirical Novelty And Significance:** 2
**Recommendation:** 6

**Clarity, Quality, Novelty And Reproducibility:**

The paper is relatively clear in writing and has extensive experiments. The method is novel. Values of hyperparameters are given for reproducibility. However, no code is provided.

**Strength And Weaknesses:**

Pros:

1, The concept of semantic scale is novel and well-motivated mathematically.

2, The usage of sphere packing method to control the scale of semantic scale is a very neat idea.

Cons:

**1, Strong correlation with model performance weakens the contribution of the proposed loss function.**  The paper highlights the correlation between semantic scale and model performance. There exits method such as hard-class mining [1] and recall loss [2] that rely on the model performance to re-weight losses. This strong correlation makes one wonder why a loss function weighted by semantic scale would be better than one weighted by performance.

**2, Adding inter-class interference seems ad-hoc and obscures the contribution of semantic scale.** In sec 3.4.1, an inter-class interference term is added to the semantic scale metric. This combination seems ad-hoc. Even though the performance of the combined metric seems to work well, it becomes less clear how much contribution the vanilla semantic scale has.

**3, Marginal effects of semantic scale and model performance w.r.t the amount of data represent correlation not causation.**  In the sec.3.3, the authors observe a strong correlation between the marginal effects of semantic scale and model performance w.r.t the amount of data. This is an interesting correlation, however, which cannot be used to explain why classification gain is marginal when data are sufficient or why performance drops quickly when data are scarce, as claimed in the abstract. Because the observation only establishes a correlation not a causation between the two events.

**4, Semantic scale is model-dependent.** Even though the authors claim that semantic scale is a natural measure that is model-dependent, it is calculated from the output (feature space) of a trained model and therefore should be model-dependent. The same dataset, with the same class statistics, will lead to different semantic scales if models are different.

[1] A. Shrivastava, A. Gupta, and R. Girshick, “Training region-based object detectors with online hard example mining,” in Proceedings
of the IEEE conference on computer vision and pattern recognition, 2016, pp. 761–769.

[2] Tian, Junjiao, et al. "Striking the Right Balance: Recall Loss for Semantic Segmentation." 2022 International Conference on Robotics and Automation (ICRA). IEEE, 2022.

**Summary Of The Paper:**

The paper proposes a new concept semantic scale, which conceptually captures the richness or diversity of the feature space, e.g., “Bird” features should be richer than “Swan” features. This quantity is quantified by the volume of the learned feature space, proportional to the determinant of feature covariance matrix. Using this new concept, the paper attempts to explain 1) the change of performance improvement w.r.t the number of training data and 2) implicit semantic bias in classification when classes are balanced. Finally, based on the discrepancy in semantic scales, the paper proposes a dynamically weighted loss function to show improvement on classification under balanced and imbalanced settings.

**Summary Of The Review:**

The paper presents a novel idea to quantify implicit bias in classification. However, questions regarding claims and some less well explained components in the method make the overall message less convincing.

---

> ### Author Response · Authors · 2022-11-13
> **Response to Reviewer kLsB (Weaknesses 1 and 2).**
>
> Dear Reviewer kLsB,
>
> &nbsp;
>
> Thank you very much for your valuable and insightful comments.
>
> First, our response to **Weakness 1** is as follows.
>
> Faced with your comments, we suddenly thought: what is the point of proposing a variety of cost-sensitive learning methods? For example, using the inverse of the number of samples \ the effective number of samples to reweight the loss, rather than directly using the accuracy of each class to weight. After careful consideration, we believe there are several reasons.
>
> **(1)** Reweighting loss with class accuracy may cause the model to over-focus on weak classes, so that other classes are ignored. Recent studies have shown that reweighting the loss strictly by the inverse of the number of samples has a modest effect [1][2]. Some "smoother" methods perform better, such as taking the square root of the number of samples [1] as the weight. [3] argues that the reason why the "smoother" method performs better is due to the existence of marginal effects. Our approach can be understood as a smoothed version of class accuracy because our proposed semantic scale has marginal effects and a high correlation with class accuracy. We note a recent work (CDB loss) published in IJCV that measures class difficulty. In addition, domain balancing also measures class-level difficulty, so we compare semantic-scale-balanced learning with them. The introduction and comparison experiments of the above two methods are shown in **Appendix H.2**.
>
> [1]Exploring the limits of weakly supervised pretraining
>
> [2]Distributed repre- sentations of words and phrases and their compositionality
>
> [3]Class-balanced loss based on effective number of samples
>
> **(2)** The method of weighting with model performance cannot bring us new cognition. Why do models perform poorly on some data and well on others? For example, face recognition models usually do not perform well in dark environments. When we encounter such a problem, the first thing to think about is whether the lack of data in the dark environment causes the model to not be fully learned. Since there is a lot of data available, this problem is not caused by the few samples, so is there any other explanation? We argue that the pattern of faces in the dark environment is not rich enough, which leads to a large number of samples clustered around the manifold with smaller volumes, making it difficult to distinguish between faces. Our approach is not only to address the performance imbalance, but also to advance researchers' understanding of deep neural networks. Advances in science are usually accompanied by the establishment of new cognition.
>
> **(3)** Our proposed semantic scale has great potential for application. In engineering applications, how many samples should be collected for each class is the most appropriate? When too few samples are collected, the class is under-represented, while too many will consume huge costs. Our approach can effectively solve this problem by stopping the collection when the semantic scales tend to be saturated. When we communicate with technology companies, we find that they have 100 million data, but there is no proper way to select representative data. So we design an idea to select representative data using semantic scales, the details of which are added in **Appendix I**.
>
> **Weakness 2:** We have weakened the effect of inter-class interference when designing the measurement of semantic scale imbalance. The Pearson correlation coefficients between class accuracy and inter-class interference, semantic scale, and semantic scale considering inter-class interference, respectively, are shown in Table 1. It can be seen that the Pearson correlation coefficient between semantic scale and class accuracy on CIFAR-10-LT without considering inter-class interference still reaches 0.8688, while the correlation coefficient between inter-class distance W and class accuracy is only 0.2957, which illustrates the importance of semantic scale. In addition, we have added the correlation coefficients between the effective sample numbers and class accuracies in Table 1. It can be observed that the correlation between effective sample number and class accuracy is almost the same as the correlation between sample number and class accuracy, which is due to the fact that effective sample number is a monotonic function of sample number. To demonstrate the performance of dynamic semantic-scale-balanced learning without considering inter-class interference, we conducted experiments on ImageNet-LT and supplemented them in **Appendix H.1**. It can be observed that dynamic semantic-scale-balanced learning without considering inter-class interference still has a strong performance.
>
> &nbsp;
>
> Yours sincerely,
>
> Authors of Paper 42

---

> ### Author Response · Authors · 2022-11-13
> **Response to Reviewer kLsB (Weaknesses 3 and 4).**
>
> Dear Reviewer kLsB,
>
> &nbsp;
>
> **Weakness 3:** We clarify the experiments shown in Figure 2 below. Previous studies have observed that **(1)** given sufficient data, the classification performance gain is marginal with additional samples. **(2)** When the data is insufficient, the classification performance drops sharply as the number of training samples decreases. We speculate that phenomenon 1 may be caused by the marginal effect of feature diversity. It should be noted that CB loss considers marginal effects, but it only qualitatively describes the gradual flattening of feature diversity with the increasing number of samples. Taking CIFAR-10 as an example, we first select a few samples for each class, train the model and test the accuracy. Then new samples are continuously added to the original samples instead of re-selecting more samples to train the model. While increasing the data we find that there are marginal effects of semantic scale, which indicates that our proposed measurement is as expected. The marginal effects of feature diversity explain phenomenon 1.
>
> However, phenomenon 2 is not explained by the marginal effects, and the effective number of samples from CB loss does not predict phenomenon 2 at all, because the effective number of samples does not grow faster than the number of samples (which we have analyzed in Section 2). We experimentally find that when the samples are few, the feature diversity measured by the semantic scale increases rapidly with the number of samples, and this increase is faster than the linear increase. The rapid increase of feature diversity measured by the semantic scale explains phenomenon 2.
>
> **Weakness 4:** The measurement of semantic scale is mathematically model-independent and it can be applied to image space as well as to speech data and other types of data. In this work, we have utilized the proposed measure in both image space and feature space. The feature distribution obeys the manifold distribution law, which means that the features of each class are distributed around a low-dimensional manifold. The classification task can be considered as the unwinding and separation of feature manifolds, and then the geometric characteristics of the feature manifolds may affect the model bias. We wish to explore the effect of intrinsic factors of the data distribution on the model bias, rather than just re-weighting the loss. Future work is discussed in **Appendix J**, which includes how to introduce semantic scale into the field of object detection.
>
> Thank you very much for your review, we look forward to hearing from you and wish you happy work.
>
> &nbsp;
>
> Yours sincerely,
>
> Authors of Paper 42

---

### Official Review · Reviewer_MKbH · 2022-10-21

**Confidence:** 5
**Correctness:** 3
**Technical Novelty And Significance:** 4
**Empirical Novelty And Significance:** 4
**Recommendation:** 8

**Clarity, Quality, Novelty And Reproducibility:**

The paper is now very clear. The evaluation is more extensive and convincing. I find the approach & contribution very original.

**Strength And Weaknesses:**

Strengths:
- I am very fond of the idea of measuring imbalance and using such measures instead of counting the number of samples in classes. This is a very promising research direction.

- I am very fond of the idea of extending the mechanism of Cui et al. [10] to develop a measure that better reflects imbalance among classes.

- I am very fond of the analysis and the insight provided by the paper, before comparing it with other methods.

Weaknesses:

Although I am very fond of the approach and the ideas, I believe that the paper has significant limitations and is not ready for publication yet. Here are the major issues that I've identified, the minor problems are listed further below:

1. First of all, the overall story of the paper can be improved:

1.1. The paper starts with a very convincing example of semantic scale (birds and swans), motivating the overall approach of introducing a measure for it, then it does not provide any insight or analysis about whether or not the proposed measure is really capturing it: The analyses in Section 3 do not provide any information about the class labels nor a discussion about them.

1.2 Although the motivation and the story of the paper is toward imbalance in classification problems, in the experimental analysis section, the paper adopts a deep metric learning setup and compares the proposed approach against the SOTA method of that domain. Although an analysis in a deep metric learning setup is not irrelevant, the story of the paper necessitates performing an extensive comparison with the SOTA of imbalance mitigation strategies.

2. I recommend comparing the proposed scale against other measures of hardness / difficulty. I find the lack of such an analysis to be a significant limitation of the paper.


Typos and Unclear Bits:
- "As the information of data increases" => What's meant is unclear.
- Section 3.2, 5th line: Error in notation. The covariance matrix for vector z_i is being defined, yet the summation runs over index i. I recommend using j in the summation just to avoid confusion.
- Eq. 1: Did you want to write "Vol(z_i)" instead, as implied by the previous sentence? Or you should update the sentence.
- "Considering that real-world metrics ... scales. We" => "scales, we". Otherwise, the sentence "Considering that..." is incomplete.
- Figure 2 is way too small and not readable.
- "which indicates that the quantitative measurement of semantic scale is reasonable. In addition, the growth rate of the semantic scale varies across classes, which is determined by the grain size of the class itself. It leads to different semantic scales even if all classes have the same number of samples." => From Figure 2, what we see is that (i) semantic scale saturates and (ii) it is different for different classes. I would describe these results as "as expected" and consider "reasonable" a strong word.
- Figure 2 right column: Changing dataset size requires re-tuning the hyper-parameters. Without re-tuning the hyper-parameters, it is expected that the performance will drop when data is subsampled. It is not clear whether this has been performed and whether the results in Figure 2 are reliable.
- "the centers of all classes are C" => It is not clear in what space these centers are.
- "C" is used in Section 3.3 and 3.4 for two different things. It is better to use different symbols. There is a similar problem with N (there might be other symbols).
- Table 1: It would have been nicer to add E_n to the table.
- Section 3.4.1: It would be nice to visualize or provide insight about the calculated weights.
- Section 5.1 and "We create long-tailed CIFAR-100 and long-tailed Cars196 for training using the first 60 and 98 classes of CIFAR-100 and Cars196, respectively, and the test sets are the remaining classes." => The authors should cite the literature for the setup.

**Summary Of The Paper:**

In this paper, the authors study the problem of imbalance in classification problems. To be specific, they propose a measure to quantify the `semantic scale' of classes and use this measure as a means to analyze the imbalance among classes. Moreover, they use this measure to mitigate the effect of imbalance in various classification datasets. They report significant improvements over the baseline methods.

I have reviewed an earlier version of the paper for ICML2022 and I see that the authors have significantly improved their paper since then (e.g. experiments on ImageNet, iNaturalist, MNIST & MNIST-LT). Thank you for taking into account my earlier comments.

**Summary Of The Review:**

Fond of the paper. Novel approach, good analyses & insights, strong results.

---

> ### Author Response · Authors · 2022-11-13
> **Response to Reviewer MKbH.**
>
> Dear Reviewer MKbH,
>
> &nbsp;
>
> We sincerely thank you very much for your valuable and constructive comments, which make our work even better.
>
> **Weakness 1.1:** Your concern is whether our proposed measurement of semantic scale can accurately measure the feature diversity of a class while considering the hierarchical relationship of semantic concepts. The nesting relationship of semantic concepts corresponds to the nesting relationship of feature manifolds, which ensures that the hierarchical relationship (i.e., proportionality) of label semantics can be satisfied when the volume of sub-manifolds is used to measure feature diversity. We present the semantic scale of multiple Stanford manifolds with different volumes in **Appendix D.1**. It can be seen that as the point cloud manifold is scaled up, the calculated volume then increases slowly and monotonically and is numerically stable, indicating that our approach can accurately measure the relative size of the manifold volume and that the numerical stability helps mitigate the effect of noisy samples. Following your previous suggestion, we have shown all class labels on CIFAR-10/CIFAR-10-LT and discussed the experimental results in **Appendix D.3**.
>
> **Weakness 1.2:** Based on your comments, we conducted experiments on ImageNet-LT and iNaturalist2018, benchmark datasets for long-tailed classification, as well as on a balanced dataset.
>
> **Weakness 2:** We note a recent work (CDB loss) published in IJCV that measures class difficulty. In addition, domain balancing also measures class-level difficulty, so we compare semantic-scale-balanced learning with them. The introduction and comparison experiments of the above two methods are shown in **Appendix H.2**.
>
> **Typos and Unclear Bits:**
>
> 1. "As the information of data increases" means that as the information contained in the dataset increases, it can also be thought of as an increase in feature diversity.
>
> 2. The covariance matrix and the volume of the manifold are calculated with the data matrix Z. The space cannot be spanned by a single vector, so Vol(Z) does not need to be modified.
>
> 3. All the symbolic problems have been revised in the newly submitted paper.
>
> 4. We enlarged and explained Figure 2 in detail in **Appendix K**. The experiments corresponding to each point in Figure 2 are trained from scratch.
>
> 5. We supplemented the Pearson correlation coefficients between the effective number of samples E_n and the class accuracy in Table 1.
>
> 6. We supplemented the citations of the literature.
>
> **Additional response:** We show more experimental results and analysis in **Appendix H**. Potential applications of our approach to other problems are presented in **Appendix I**. Future research directions and how to introduce semantic scale imbalance into object detection are discussed in **Appendix J**.
>
> Thank you very much for your review, we look forward to hearing from you and wish you happy work.
>
> &nbsp;
>
> Yours sincerely,
>
> Authors of Paper 42

---

> > ### Comment · Reviewer_MKbH · 2022-11-16
> > **Thank you**
> >
> > Dear authors,
> >
> > Thank you for the detailed responses. I am in general happy with your responses but Figure 6 in the Appendix is only demonstrating that an increase in volume increases your measure. It does not demonstrate anything about semantic scale.
> >
> > Best

---

> > > ### Author Response · Authors · 2022-11-17
> > > **Additional Responses to Reviewer MKbH**
> > >
> > > Dear Reviewer MKbH,
> > >
> > > &nbsp;
> > >
> > > Thank you very much for your reply.
> > >
> > > In response to your question, we propose a matching method of semantic hierarchy in **Appendix L** and explore whether the semantic scale can capture the semantic hierarchy of the dataset. The experimental results are presented in **Table 12**, and it can be seen that we successfully matched each child class to the parent class. You can view the details by **downloading the latest paper**.
> > >
> > > If you have questions, you can feel free to contact us.
> > >
> > > &nbsp;
> > >
> > > Yours sincerely,
> > >
> > > Authors of Paper 42

---

### Author Response · Authors · 2022-11-13
**Response to all reviewers.**

Dear Reviewers,

&nbsp;

We are very grateful to the reviewers for their insightful and constructive comments, and we have summarized all comments and added responses to the following four questions in **Appendix H**. In addition to the supplemental appendices, revisions or additions in the main text have been highlighted in **red** for ease of retrieval. **You can view the latest version of our paper by downloading it**.

**1**. The effectiveness of dynamic semantic-scale-balanced learning without considering inter-class interference.

**2**. Comparison with other methods of measuring class-level difficulty.

**3**. The performance of dynamic semantic-scale-balanced learning in generalized long-tailed learning.

**4**. Dividing ImageNet into three subsets based on semantic scale, and showing the performance of dynamic semantic-scale-balanced learning on the three subsets.

We supplement the idea of utilizing semantic scale to address other problems in **Appendix I**, and discuss future work in **Appendix J**.

&nbsp;

Yours sincerely,

Authors of Paper 42

---

### Author Response · Authors · 2022-11-17
**Please feel free to contact us.**

Dear Reviewers,

&nbsp;

We answered each reviewer's questions and believe that the reviewer's concerns have been addressed. If you have any questions or concerns, we hope to help you answer them.

In addition, we propose a matching method of semantic hierarchy in **Appendix L** and explore whether the semantic scale can capture the semantic hierarchy of the dataset. The experimental results show that we succeeded in matching each child class to the parent class, and you can check it out by **downloading the latest paper**.

We look forward to hearing from you, so please feel free to contact us.

&nbsp;

Yours sincerely,

Authors of Paper 42

---

### Public Comment · ~Feng_Hong1 · 2023-02-28
**Congratulations and a request for the code.**

Hi,

Nice work and congratulations! I am very interested in your proposed method. Is there any plan for open source code?

Best,

Feng Hong

---

> ### Author Response · Authors · 2023-03-02
> **Thank you for your attention!**
>
> Dear Feng Hong,
>
> Thanks for your interest, we plan to open source a geometry toolkit for data manifolds that will be combined with our other work, stay tuned, but this may take some time.
>
> Best,
> Yanbiao Ma

---

> > ### Public Comment · ~Feng_Hong1 · 2023-03-08
> > **Some questions on implementation details**
> >
> > 1. Equation 3. Why does Equation 3 not contain $\epsilon$, since $\epsilon$ is set to 1000 in the previous paragraph? Here I want to check whether Equation 3 contains $\epsilon$ and why $\epsilon$ is set so large.
> > 2. After calculating $S=S'\odot W$, is further normalization required?
> > 3. Maybe I missed it, but I didn't find the size of the storage pool $Q$.
> >
> > Many thanks

---

### Decision · Program_Chairs · 2023-01-20

**Decision:**

Accept: poster

**Justification For Why Not Higher Score:**

I find there are quite a few details that are not very clear, particularly with regards to Equation 3.

**Justification For Why Not Lower Score:**

Good empirical results, reviewers consistently appreciated the overall contribution.

**Metareview: Summary, Strengths And Weaknesses:**

The paper proposes a notion of semantic scale, which aims to capture more general notions of imbalance beyond counting samples per class. It is shown that this notion has some interesting properties, such as having a saturating behavior, and that it can be used to reweight samples so as to improve performance on long-tailed and regular settings.

While the final reviewer scores are all positive, my own assessment is more qualified, and closer to the original reviewer scores. The core idea of trying to measure a more refined notion of imbalance, rather than counting number of samples in a class, is certainly well motivated and interesting. The basic idea of measuring volumes in the feature space is also sensible in abstract. However, I find there are quite a few details that are not very clear. In particular, Equation 3, which appears the final definition of semantic scale, is not (in my reading) clearly derived. Further, many details of the DSB loss are relegated to the Appendix, although DSB is claimed as one of the main contributions. A few of these points might be excusable, and fixable with minor edits and clarification. However, they cumulatively make the paper hard to follow on a technical level.

While we uphold the overall reviewer consensus, the paper could be strengthened by greater clarity on several points:
- "Assume that the volume of each sample is unit volume 1" -> not clear. How does a single sample have a volume?
- "As the information of data increases, the probability P will be higher." -> not clear. What is "information of data"? Why should more "informative" data have more overlapping samples? The response mentions: "as the information contained in the dataset increases, it can also be thought of as an increase in feature diversity". This is far too vague.
- "which indicates that the effective number of samples does not increase faster than the number of samples, but this is not the case in our experimental results" -> confusing. Do you mean the theoretical result does not hold in practice? Or do you mean you use some other notion of effective number of samples? Or something else?
- "a feature mapping function f(x, θ) and a trained downstream classifier g(z), i.e., x → z(θ) → y" -> not clear. What is z(θ)? The next equation suggests it is f(x, θ). How does this relate to g(z)? Does z(θ) → y mean that you train a classifier g(.) that implements a mapping from z(θ) to y?
- "After the determinant expansion, the characteristic polynomial..." -> confusing. You can just state the standard fact that the product of eigenvalues is the determinant.
- "The feature volume is positively correlated with the number of spheres" -> not clear. What are these spheres? Are these covering the sample?
- "and since the sphere packing has error at the subspace boundary" -> not clear what this means.
- "the error of the finite feature vectors is assumed to be independent additive Gaussian noise" -> what is this error, exactly? Is it the inherent uncertainty in the samples? Why is noise in the learned features, rather than the inputs, appropriate?
- "Nε=Vol(Z′)/Vol(w)" -> what is w? How does it differ from w_i?
- "The dimension of feature zi is d, so let n = d" -> not clear at all why this assumption is justified. Isn't n the total number of training samples?
- "In practical training, it is essential to normalize the feature vectors so that their mean value is 0" -> I am not sure on this point. In, say, ResNets, to my knowledge the final learned embeddings are not zero mean.
- It is not clear why (1) and (3) both refer to the volume of Z on the LHS, but have very different expressions.
- "How to combine general loss to generate DSB loss is described in Appendix F.1." -> This is not appropriate. If this is meant to be claimed as one of the technical contributions, the precise scheme should be summarized in the body.
- "because both the semantic scale and the number of samples are natural measures and they are not model-dependent" -> not clear why this establishes the previous assertion.
- "our proposed semantic scale, like the number of samples, is a natural measure of class imbalance and does not depend on the model’s predictions" -> I don't understand this claim. Doesn't it depend on the model's learned embeddings? Is that really very different than basing it on the learned logits/predictions, which are just an additional layer on top of the embeddings?

**Note From Pc:**

if the above contains the word "oral" or "spotlight" please see: "oral" presentation means -> notable-top-5% and "spotlight" means -> notable-top-25%. As stated in our emails, we are disassociating presentation type from AC recommendations

**Summary Of Ac-Reviewer Meeting:**

See summary post below. Most reviewers found the basic idea interesting, but there was lingering uncertainty about the justification of semantic scale versus accuracy. These were followed up offline.

_Attendees_

AC, Reviewers MKbH, kLsB, bNtG

We could not find a time-slot suitable for Reviewer DeDZ, due to the different locations of all participants

_Meeting notes_

Reviewer MKbH: initial review found some flaws with the presentation. However, after the response and revision, satisfied with the paper. The idea of going beyond counting the number of samples as a measure of imbalance seems pretty interesting, and is an important long-term direction for the field.

Reviewer kLsB: agreed that a more refined notion of imbalance is appealing. Main concerns are that inter-class interference is somewhat ad-hoc and dilutes the message; and that the central claim of the importance of semantic scale relies on a correlation (as opposed to causation).

Reviewer bNtG: agreed with kLsB that the inter-class interference point can be confusing. Main concern was that the paper reads more like a mix of several ideas. However, the response clarified some details, and added some new results. Overall, inclined to weakly accept the paper.

Reviewer MKbH: inter-class interference is a natural idea, so it is a good contribution to the field to quantify the effect it has; saves other researchers from going down this path!

Reviewer kLsB: still not sure about the central claim of the importance of semantic scale. This is based on showing that there is a strong relationship between semantic scale and accuracy. However, if this is the case, then why not just use the accuracy as the central measure? Also, the experimental results are sometimes a bit mixed, e.g., if we compare the existing LDAM versus the proposed CE + DSB.